# Topology optimization on metamaterial cells for replacement possibility in non-pneumatic tire and the capability of 3D-printing

**Shokouh Dezianian**[1], **Mohammad Azadi**[1]*, **Seyed Mohammad Javad Razavi**[2]

**1** Faculty of Mechanical Engineering, Semnan University, Semnan, Iran, **2** Department of Mechanical and Industrial Engineering, Norwegian University of Science and Technology, Trondheim, Norway

* m_azadi@semnan.ac.ir

**Data Availability Statement:** All data that support the findings of this study are available inside the paper based on tables and figures.

## Abstract

One of the applications of mechanical metamaterials is in car tires, as a non-pneumatic tire (NPT). Therefore, to find a suitable cell to replace the pneumatic part of the tire, three different solution methods were used, including topology optimization of the cubic unit cell, cylindrical unit cell, and fatigue testing cylindrical sample (FTCS). First, to find the mechanical properties, a tensile test was conducted for materials made of polylactic acid (PLA) and then, the optimization was done based on the weight and overhang control for the possibility of manufacturing with 3D printers, as constraints, besides, the objective of minimum compliance. In the optimization of the cubic unit cell, the sample with a minimum remaining weight of 35% was selected as the optimal sample. However, for the cylindrical unit cell, a sample with a weight limit of 20% was the most optimal state. In contrast, in the FTCS optimization, a specimen with lower remaining weight equal to 60% of the initial weight was selected. After obtaining the answer, five cells in the FTCS and two mentioned cells were evaluated under compressive testing. The samples were also subjected to bending fatigue loadings. The results demonstrated that cellular structures with 15% of lower weight than the optimized samples had the same fatigue lifetime. In the compressive test, the line slope of the specimens with cellular structures in the elastic region of the force-displacement diagram was reduced by 37%, compared to the completely solid samples. However, the weight of these samples decreased by 59%. Furthermore, the fracture surface was also investigated by field-emission scanning electron microscopy. It was observed that a weak connection between the layers was the cause of failure.

## 1) Introduction

Metamaterials have properties beyond natural materials. However, we can name limited examples of metamaterials in nature, including cork with a Poisson ratio close to zero [1]. The properties of metamaterial are depending not only on the base material but also on the architecture of the structure. Metamaterials are produced from the repetition cells together. Recent studies show that cell structures can be designed in such a way that mechanical characteristics such as

**Funding:** This work is funded by Iran National Science Foundation (INSF) under project No. 4002601. Moreover, the funders had no role in study design, data collection and analysis, decision to publish, or preparation of the manuscript.

**Abbreviations:** AMM, Acoustic Metamaterials; CS, Cell Structure; DLP, Different Load Point; EBM, Electron Beam Melting; FFF, Fused Deposition Modeling; FE-SEM, Field-Emission Scanning Electron Microscopy; FTCS, Fatigue Testing Cylindrical Sample; HC, Hollow Cubic (unit cell); MMM, Mechanical Metamaterial; NPT, Non-pneumatic tire; PLA, Polylactic Acid; RD, Rhombic Dodecahedron; SLA, Stereolithography; SLM, Selective Laser Melting; SLS, Selective Laser Sintering; TC, Truncated Cuboctahedron; TMM, Thermal Metamaterial; TPMS, Triply Periodic Minimal Surfaces; TPU, Thermoplastic Polyurethane; TO, Topology Optimization.

lightweight, vibration control, high-energy absorption, and advanced thermal performance could be obtained [2]. In general, cellular structures can be divided into three categories of (1) foams, (2) lattice, and (3) Triply Periodic Minimal Surfaces (TPMS), which include solid-based and sheet-based TPMs [3]. Metamaterials are also divided into four categories including Electromagnetic Metamaterial (EM), Acoustic Metamaterials (AMM), Mechanical Metamaterial (MMM), and Thermal Metamaterial (TMM) [4]. One of the applications of mechanical metamaterials is in car tires. In recent decades, researchers have replaced the air in tires with the use of metamaterial cells and designed Non-Pneumatic Tires (NPT). NPTs generally consist of four components: core, cell, shear layer, and tread [5–7].

One of the advantages of NPTs is the reduction of rolling resistance [6–9]. In addition, the most obvious feature of this type of tire is the elimination of the concern for puncture and the need to adjust the air pressure inside the tire [7]. Moreover, due to the mesh structure and lack of sidewalls in NPTs, this type of tire uses fewer primary raw materials and is therefore more compatible with the environment [8,10].

Another advantage of these tires is improving the comfort of passengers due to better shock absorption [7]. Since there is no need for an air tube in NPTs, some designs provide more available space inside the tire for brakes or even electric motors. NPTs in general have higher stability and longevity compared to traditional tires and their fabrication via 3D printing could result in energy and cost saving [8,9].

NPTs have a wide range of potential applications among which their use in wheelchairs, space probes, cars, trucks, bicycles, can be pointed out [11]. Various architectures of NPTs have been designed and studied in the past, including Tweel [12–23] and Mechanical Elastic Wheel (MEW) [24–31] tires, which are extensively researched and presented in S1 File.

Mathew et al. [32] introduced a model of a NPT made of natural rubber materials instead of synthetic rubber in the tread and polyester instead of nylon in the wall. In addition, these researchers studied different patterns. The study showed that the tire with diamond structure with artificial materials had lower deformation than other structures. Zhang et al. [33] designed a NPT according to the lower body of a kangaroo. By comparing a pneumatic tire with the same size, a radial stiffness comparison test was performed to ensure the accuracy of the numerical analysis. Radial hardness, lateral hardness, longitudinal hardness, torsional hardness and pressure in the contact area under different loads were studied. The results illustrated that the NPT had a better performance than the pneumatic tire. Zhang et al. [34], in another research, fabricated three types of NPTs with different patterns and compared the dynamic and stable performance of these tires with pneumatic tires.

Wang et al. [35] used thermoplastic polyurethane (TPU) to fabricated NPTs with the fused filament fabrication (FFF) technology. The results demonstrated that the optimal nozzle temperature for 3D printing of the selected TPU materials was 210˚C. It was also determined that FFF technology could be a suitable method for producing TPU NPTs. Papageorgiou et al. [36] investigated the effect of a range of geometrical parameters on the mechanical behavior and weight of NPTs with a honeycomb structure. Therefore, a finite element model of the NPT under vertical loading was designed to evaluate the maximum stress, maximum vertical displacement, contact pressure, and absorbed energy during loading. Zheng et al. [37] studied the effect of different angles in the honeycomb pattern on the multiaxial stiffness and tire characteristics in the contact area.

Mazur [9] compared a NPT with pneumatic tires. It was concluded that experimental results did not contradict the claims of manufacturers that NPT had higher damping properties while having lower rolling resistance. Fu et al. [38] studied the factors affecting the fatigue lifetime of NPTs. In that study, the analysis of the effect of structural parameters on the fatigue lifetime, the prediction of the fatigue lifetime of NPTs, and the evaluation of the fatigue

performance of these tires were investigated. Using the numerical simulation approach, the effect of parameters such as angles, curvature, tread thickness, and thickness of side cells on tire fatigue lifetime were evaluated. Ghasemi et al. [11] presented an analytical model for a NPT. Deformations caused by cutting and bending were considered in the model. A good agreement of the model was seen with the experimental data of rolling resistance by comparing with two computational models utilizing ABAQUS.

Ju et al. [5] introduced a NPT with a compliant cellular solid spoke component. In that work, hexagonal honeycomb spokes for a high fatigue resistance design were investigated by seeking compliant hexagonal structures that had low local stresses under macroscopic uniaxial loading. Using the honeycomb mechanics, two cases of hexagonal honeycombs were designed, (1) the same load-carrying capacity and (2) the same cell wall thickness. The results demonstrated that the hexagonal honeycombs with a highly positive cell angle had low local stresses and low mass under the same vertical load carrying capability.

Maharaj and James [39] used metamaterials in the shear layer. It was designed by optimization and constraints of stress and buckling of the shear layer. The obtained cells showed favorable properties for use in NPTs. Jin et al. [6] investigated NPTs with honeycomb cells. In the static analysis, it was observed that the maximum stress in the tread and cells was lower than that of pneumatic tires. The dynamic analysis that the stress level in the tread and cells was higher than in the static state. Decreasing the angle of the cell also led to a decrease in rolling resistance.

According to the literature review, the concluding marks could be listed as follows,

- The known metamaterial cells were replaced in the tire and their mechanical properties were compared with each other [32–34].

- The shape parameters such as the cell angle were investigated in different patterns, especially the honeycomb pattern [5,6,36,37].

- The use of 3D printers to fabricate the NPT was observed in the previous articles [5,8,9].

However, in none of the above articles, the topology optimization was done to find a suitable cell. Moreover, in none of the mentioned research, properties such as compressive strength and bending fatigue lifetime of tires were not investigated. In addition, the layer orientation and its properties were not studied. Therefore, the innovation of this research could be claimed as follows,

- Finding a cell with resistance to compressive forces and bending cyclic loads (fatigue), compared to other cells and solid tires.

- Using the geometry constraint (the overhang control) in the connection with the 3D printing of NPTs, in order to have no supports for the structure.

## 2) Materials and experiments

### 2.1) Patterning of cells

To topologically optimize and design a suitable unit cell to replace the pneumatic part of the tire, a review was done from different sources. Table B1 in S2 File shows different types of cells (different design methods: Topology optimization, numerical methods, inspired by nature, random shape parameters) and their properties [40–73]. According to Table B1, Hollow Cubic (HC) unit cell, due to properties such as higher elastic modulus with lower Poisson ratio [54,66–67], higher strength against compressive forces [54,69], and longer fatigue lifetime

**Table 1. The dimensions and the layout type of the HC unit cell.**

| Ref. | Structural geometry | Dimensions of the structure (mm) | Number of cells | The radius of the cell fillet (mm) | Cell thickness (mm) | Cell length (mm) | Fabrication Process |
|---|---|---|---|---|---|---|---|
| [46] | cubic | - | 5⊆5⊆5 | - | - | - | EMB |
| [48] | Cylindrical | 15×10 | - | - | 0.3 | 1.4 | SLM |
| [44] | Cylindrical | 13.5×10 | - | 0.13 | 0.26 | 1.24 | SLM |
| [50] | cubic | - | 3⊆2⊆2 | 0.2 | 0.7 | 4 | SLM |
| [56] | Cylindrical | - | - | - | 0.65 | - | EBM |
| [59] | cubic | 10⊆10⊆20 | - | - | 0.614 | 1443 | EBM |
| [51] | Cylindrical | 13.5× 10 | - | 0.13 | 0.26 | 15 | SLM |
| [60] | cubic | - | 1 2⊆2⊆2 3⊆3⊆3 | - | 1.5 | 4 | SLM |
| [62] | Cylindrical | - | 12×11 | 0.6 | 0.67 | 4 | SLM |

[46,50,56] was chosen as the unit cell. The Poisson ratio for this pattern was reported as 0.089 to 0.190 [46].

Table 1 also shows the dimensions and number of HC unit cells used in each structure. It should be noted that in this table, the discussed cells were produced using laser-based AM, such as Electron Beam Melting (EMB) and Selective Laser Melting (SLM). These processes have higher accuracy and the ability to produce parts with very small dimensions, compared to the FFF technique. Therefore, to find the smallest dimensions of the cell that could be manufactured by the FFF, which is a non-laser process, an HC unit cell was chosen with a length of 1.4 mm and a cell wall thickness of 0.3 mm and without a fillet radius. The reason for not taking into account the radius of the fillet originates from the research results of Dallago et al. [62]. In that research, it was proven that not taking into account the fillet radius leads to an improvement in the ratio of the amplitude stress to the yield stress of the structure. In other studies, the effect of the radius of the fillet was investigated and proven. The radius of the fillet depends on the parameters of the 3D print, the direction of the 3D print, and the thickness of the strut, which in some cases led to differences between the CAD file of the designed part and the produced part [50,51].

One of the advantages of additive manufacturing is the capability to fabricate complex shapes; therefore, this method can be useful for produce of metamaterial structures. By scaling the dimensions of this structure, the construction capability was checked. Fig 1 shows the

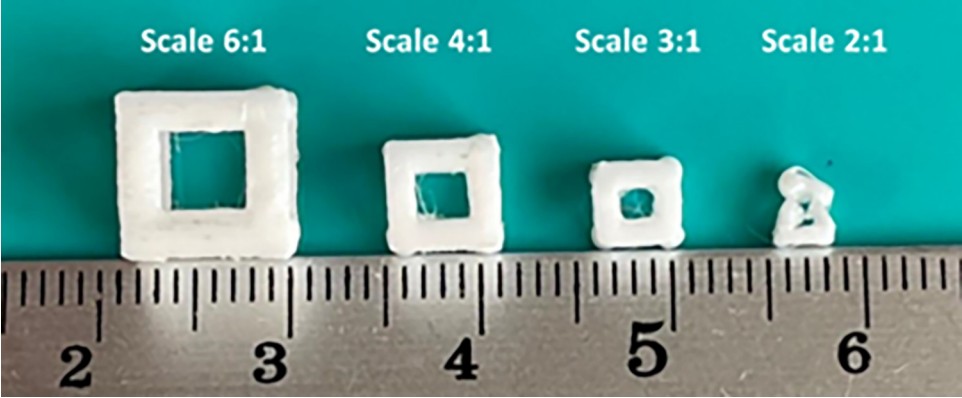

**Fig 1. The scaled samples of 3D printing.**

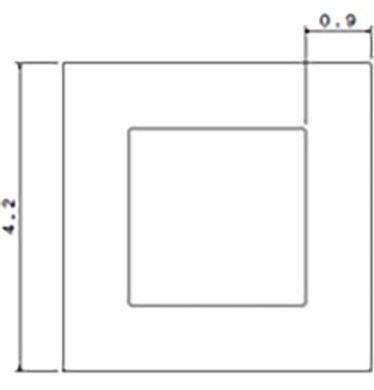
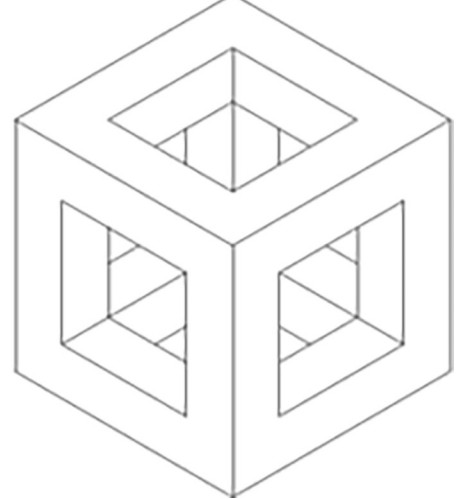

**Fig 2. The geometry and dimensions of the HC unit cell selected from the articles.**

scaled and manufactured parts. According to this figure, only cells that were at least 3 times the size of the selected cell can successfully be fabricated via FFF 3D printing. Therefore, the dimensions of the cell, used in this research, were selected according to Fig 2. Table 2 also presents the parameters of 3D printing. These parameters were selected based on their effect on mechanical properties [74–77] and the 3D printing capability of the smallest unit cell.

In addition to the HC unit cell, there were two other cells, including the Truncated Cuboctahedron (TC) due to bearing the compressive force [46] and having the longest fatigue lifetime after the HC unit cell [47,48]. The Rhombic Dodecahedron (RD) was also selected due to its mechanical properties and higher yield strength [46].

For the topology optimization, only a part of the tire was selected (not for the whole tire). After topology optimizing, the arrangement of the cells in the whole tire was done, which was discussed at the end part of this article (S6 File). Due to the choice of the HC unit cell, cubic unit cell was chosen for optimization. In addition, cylindrical unit cell was considered due to the absence of angles between lines and sharp parts, which led to a reduction in stress concentration compared to cubic unit cell.

In general, three types of geometries were selected for optimization and three types of cells for a comparison to Fig 3. In Fig 4, three types of optimizations include: (1) optimization of cubic unit cell, (2) optimization of cylindrical unit cell (after optimization, these cells can be used in a scaled form in the real tire), and (3) optimization of the Fatigue Testing Cylindrical Sample (FTCS) along with the type, the arrangement of the cells is shown.

### 2.2) Materials Specifications

The studied material was poly lactic acid (PLA). The melting temperature of PLA polymer is between 180 and 230˚C [78], its Poisson ratio is 0.36, and its density is 1240 kg/m$^3$ [79].

In this research, the tensile properties of materials, which have a more critical effect on the sample behavior, compared to compressive loads, were used. Tensile test samples were

**Table 2. 3D printing parameters in the present study.**

| Parameters | speed (mm/s) | Nozzle temperature (˚C) | Infill (%) | Layer Height (mm) | Nozzle diameter (mm) | Bed temperature (˚C) |
|---|---|---|---|---|---|---|
| Value | 20 | 180 | 100 | 0.2 | 0.2 | 25 |

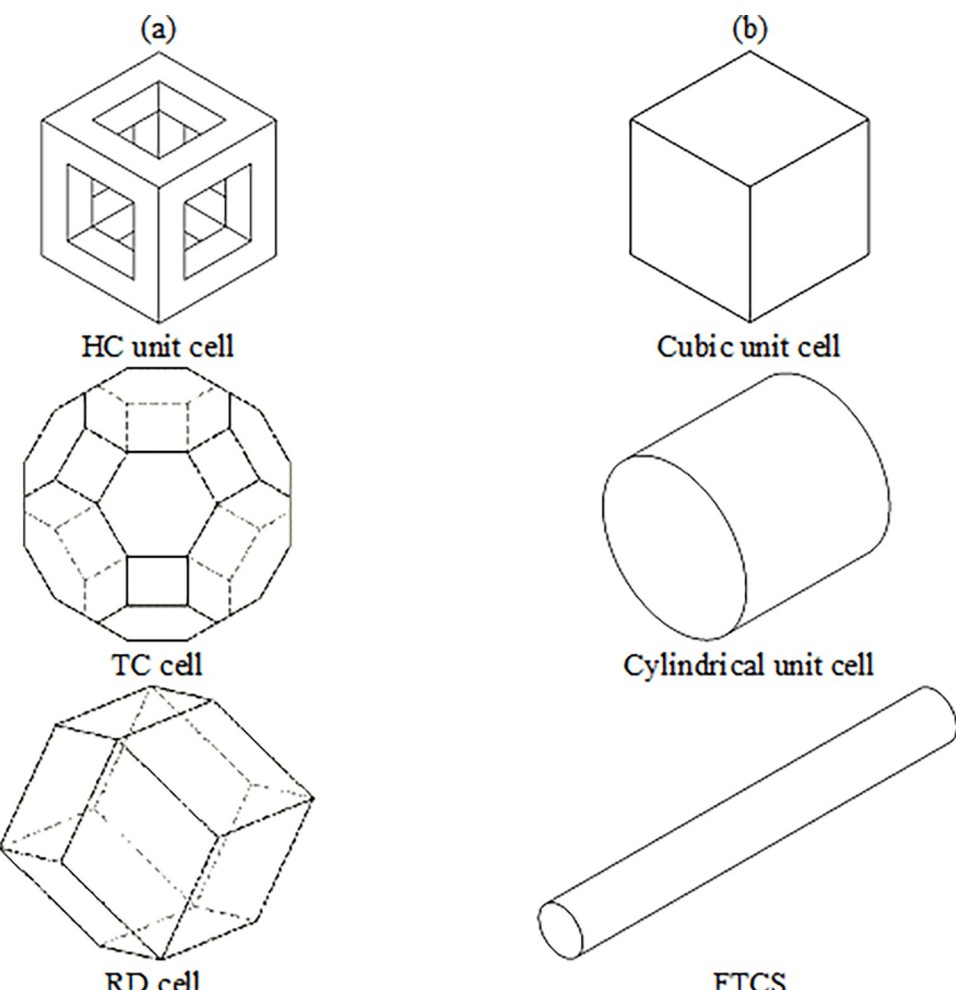

**Fig 3.** The geometry and selected cells for optimization and comparison: (a) selected cells for comparison and (b) selected geometries for optimization.

prepared according to ISO-527 standard with the FFF method and the repeatability of 3 tests and 3D printing parameters according to Table 2. Loading in this test was done at a speed of 50 mm/min. The related results are depicted in Table 3. These properties were used in the optimization of parts. The elastic modulus is the average of the slope of the linear region and the ultimate stress, the failure point stress. More information about tensile testing is provided in S3 File.

Usually, the vertical force applied to a tire is determined based on the rated load of the tire. However, in the research related to NPTs [5,6,9,11–13,17,20,22,24–39], various forces from 0.5 to 35 kN were used for different analyses. The car considered in this research was a passenger car. Therefore, considering NPTs used in vehicle (tires with small width) and comparing them with pneumatic tires, a force of 5 kN was selected as the vertical force applied to the tire.

Another hypothesis of this research was that the car moves only on a straight road without turns and at a constant speed. Therefore, lateral forces and as a result lateral sliding did not occur in the tire, but there were longitudinal forces caused by braking and acceleration. These longitudinal forces, which were caused by the friction between the tire and the road, led to a bending force in the tire. The bending force on the tire was also considered to be 5 kN. The

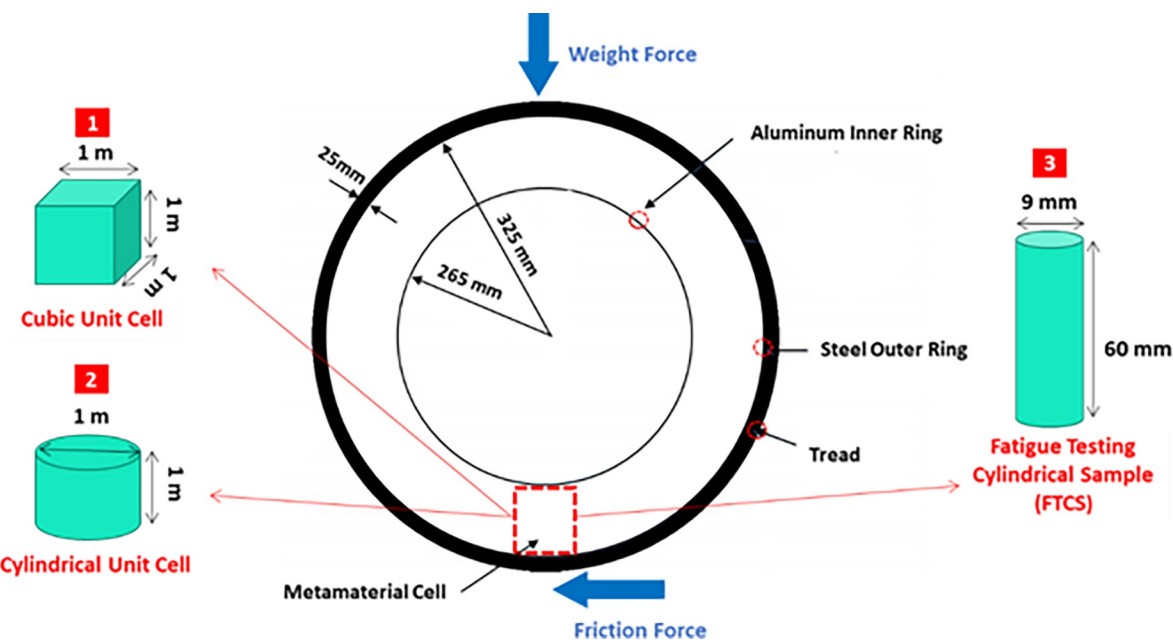

**Fig 4. Tire dimensions and the arrangement of cells.**

same values were also reported in references [80–82]. According to the obtained results, it can be said that the ratio of compressive and bending forces entering the tire was at the ratio of one to one, approximately; and considering that all the analyses were considered linearly, the force of 1 kN in the simulation was used.

## 2.3) Optimization procedures

To design metamaterials, by using the optimization method, the base cell was designed and then depending on the physical problem and the considered dimensions, the cells were placed next to each other. S4 File provides comprehensive information about topology optimization [82–86].

The optimization process was done for three types of geometry, including cubic unit cell, cylinder unit cell, and FTCS, with two compressive and bending forces of 1 kN for each type.

According the criterion proposed in the literature [87], PLA material is considered as a brittle material, which enters the elastic-plastic region under load, but the plastic region is very small. Moreover, Gigante et al. [88] proved that the PLA material was brittle. Therefore, the elastic properties of materials and linear analyses can be used in optimization processes.

It should be noted that in linear finite element modeling, the change in material properties in the elastic region does not change the von-Mises stress distribution in the structure. Therefore, there is no change in the final optimization response. In other words, the stress is dependent on the geometry of the part and independent of the material properties [89].

Optimization was done using the Tosca module in ABAQUSE software version 2021. The ABAQUS software provides the possibility of selecting different objective functions along with

**Table 3. Mechanical properties of the studied materials.**

| Filament | Ultimate Stress (MPa) | Yield Stress (MPa) | Elastic Modulus (MPa) |
|---|---|---|---|
| PLA | 55.1±1.6 | 56.0±2.5 | 3089.3±100.7 |

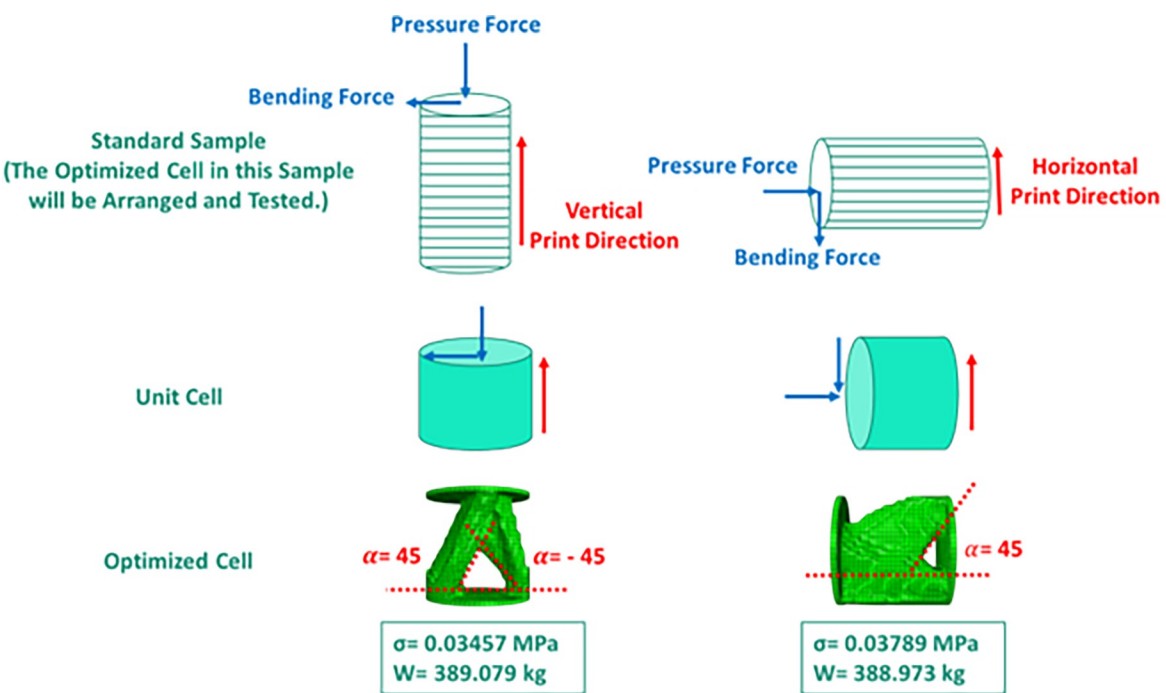

**Fig 5. The effect of print direction and overhang control on the optimization response.**

various geometric restrictions and constraints. Therefore, choosing the appropriate objective function and constraints is of particular importance.

In pneumatic tires, the air inside the tire and the tire wall have the capability to resistant the forces on the tire. However, in NPTs, there is no air or tire wall. Therefore, the desired geometry and material must be able to resistant these forces. In addition, in the other articles [89,90], the objective function was considered as a minimum compliance, compared to this work.

The volume was also a constraint in the problem. The compliance indicated by the symbol $C$ was calculated according to Eq (1) [91]. It should be noted that the unit of measurement of compliance is [1/Pa].

$$C = U^T F = U^T KU \tag{1}$$

In this relationship, $U$ is the displacement vector, $K$ is the stiffness and $F$ is the force vector. On the other hand, the strain energy was equal to the area under the displacement-force curve. As a result, compliance was related to strain energy. Therefore, since compliance does not exist directly in the ABAQUS software, the minimum strain energy was used as the objective function.

The printing orientation affects the mechanical properties of the material. This note is the first issue that is also mentioned in the literature [74–77]. In addition, the printing orientation

**Table 4. The optimization parameters in this study.**

| Geometric Restriction | Overhang control and Frozen area |
|---|---|
| Constraint | Remaining weight for 20, 40, 60, and 80% of the total weight |
| Objective Function | Minimum strain energy (or minimum compliance) |
| Design Response | Strain energy and Weight |

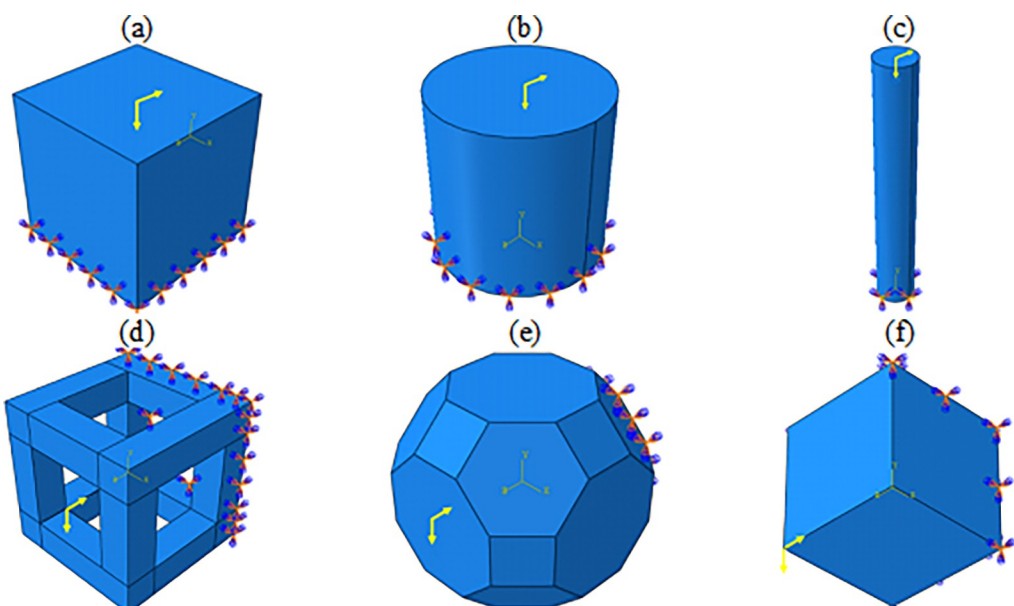

**Fig 6.** Boundary conditions for (a) cubic unit cell, (b) cylindrical unit cell, (c) FTCS, (d) HC cell, (e) TC cell, and (f) RD cell.

can affect that either the support is required or not. Therefore, it is very important to check the printing direction. This issue is due to the disadvantages of FFF 3D printers, which needs to have the support during fabrication. In fact, in parts where the angle between the part and the horizontal line is less than a certain value, the supporting structures prevent material excessive deviation from the design geometry [92]. Considering that separating these supporting structures from the main piece is a difficult task and sometimes leads to damage to the surface of the sample, in the optimization process, overhang constraints are used to create angles greater than 45˚. To apply this stipulation, it is necessary to specify the direction of printing and stacking. According to previous studies [74,75,93] on the effect of printing direction, the horizontal printing resulted in higher bending fatigue lifetime of the printed material. However, it was proven [83,94] that the vertical print direction had better compressive properties. Fig 5 shows the difference between the print direction and the overhang control on the final response, including the von-Mises stress and the weight after optimization.

The parameters used in optimization are according to Table 4. The desired geometries for optimization, the cells selected for comparison, and their boundary conditions are shown in Fig 6. Their von-Mises stress contour can be seen in Fig 7 under the combined bending and compressive load of 1 kN. The mesh convergence was checked for these geometries and the mesh size 50 mm for the cubic unit cell, 40 mm for the cylindrical unit cell and 1 mm for the FTCS were chosen. For HC, TC, and RD cells, mesh sizes of 30, 50 and 40 mm were selected, respectively.

## 2.4) Fatigue and compressive testing

In this research, a rotational bending fatigue machine with the trade name SFT-600 was used. The sample was subjected to fully reversed bending loading under stress-control conditions. In this case, the ratio of the maximum stress to the minimum stress was -1. All the fatigue tests were performed with a loading frequency of 100 Hz and at room temperature. The literature [76,77] also used the same frequency value for polymers. In addition, considering the real-life performance condition of NPTs, a high loading frequency was applied to the component.

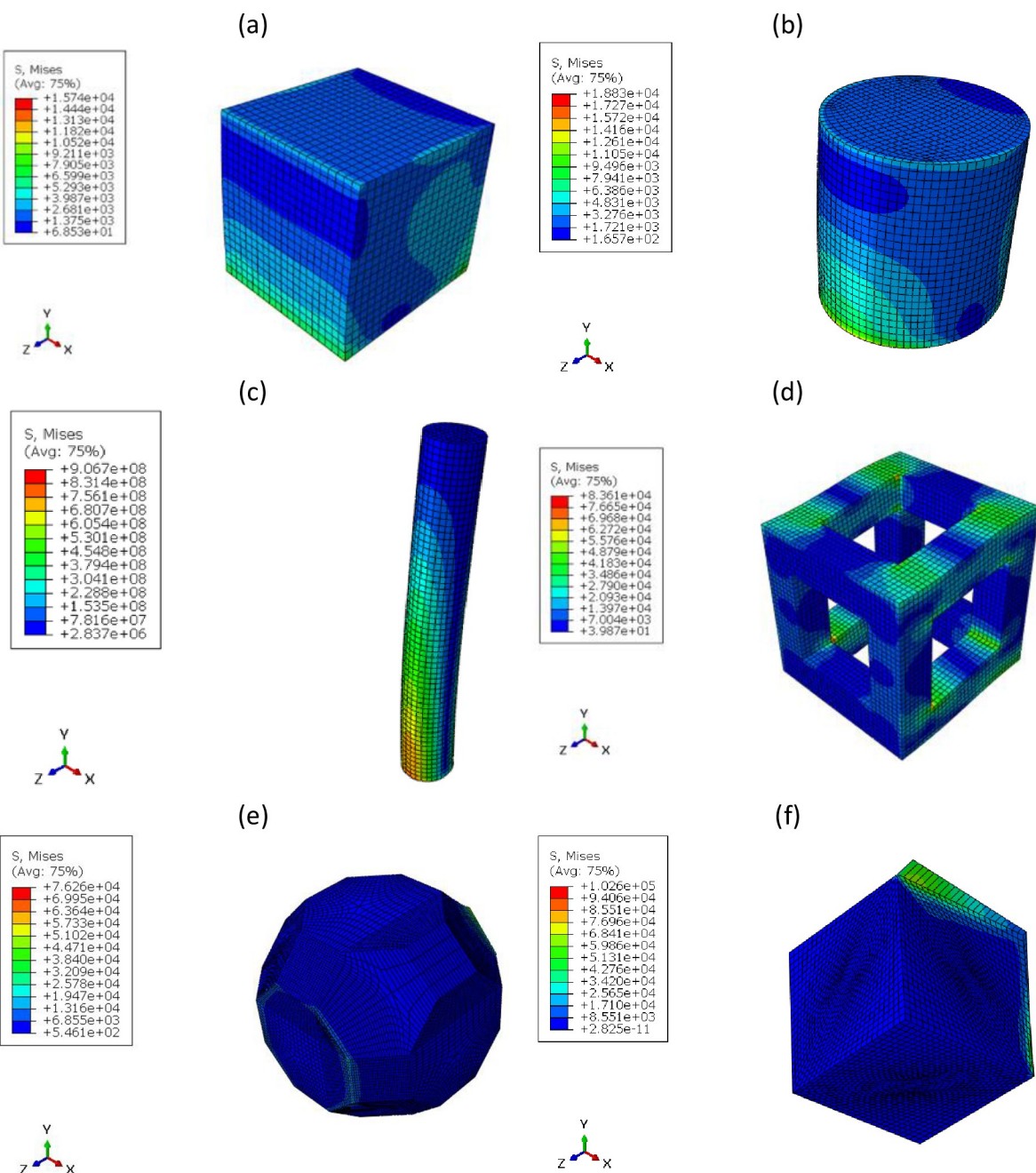

**Fig 7.** The von-Mises stress contour under 1 kN bending and compressive load for (a) cubic unit cell, (b) cylindrical unit cell, (c) FTCS, (d) HC cell, (e) TC cell, and (f) RD cell.

According to the ISO1143 standard and according to the conditions of the device, the test samples were selected as cylinders with a diameter of 9 mm and a length of 60 mm [95]. The compressive test specimen according to the ASTM D695 standard should be in the form of a cylinder whose length is lower than twice its diameter. Therefore, a diameter of 9 mm and a height of 16 mm were used in this research. The loading rate according to the standard was 1.3 mm/s. The image of the devices, used in this work, is shown in S5 File.

**Table 5. The results of cubic unit cell optimization.**

| Cell | Weight constraint (%) | Iteration* | Max. Stress (Pa) | Final weight (kg) |
|---|---|---|---|---|
| Basic geometry | - | - | $1.50 \times 10^4$ | 1240.000 |
| 1 | ≥80 | 17 | $1.88 \times 10^4$ | 988.659 |
| 2 | ≥60 | 13 | $2.80 \times 10^4$ | 732.547 |
| 3 | ≥40 | 10 | $5.00 \times 10^4$ | 486.543 |
| 4 | ≥35 | 8 | $6.30 \times 10^4$ | 427.100 |
| 5 | ≥30 | 7 | $8.00 \times 10^4$ | 366.231 |
| 6 | ≥25 | 6 | $8.80 \times 10^4$ | 304.780 |
| 7 | ≥20 | 6 | $1.03 \times 10^5$ | 243.838 |

*Repeat the solution steps until reaching the final answer.

## 2.5) Fractography analysis

To check the fracture surface of the printed parts, the surfaces were seen using a field-emission scanning electron microscopy (FE-SEM), Zeiss Sigma 300 HV, Germany. To prepare the samples for fractography analyses, a gold coating was applied on the surfaces.

## 3) Results and discussion

### 3.1) Optimization of cubic/cylindrical unit cell

The results of optimizing the cubic unit cell with the objective function of minimum compliance of PLA with dimensions of 1 x 1 x 1 m under a concentrated compressive and bending force of 1 kN are presented in Table 5 and Fig 8 and the stress contours of optimized structures are presented in Fig 9. Moreover, to make a better comparison between the optimized cells and the cells selected from the literatures, the percentage changes in the weight and tension of the cells compared to the initial solid geometry are presented in Table 6.

According to Table 6, the cells obtained from the optimization of the cubic unit cell had lower stress than the cells selected from the articles of relatively the same weight. Optimization was done for four different weights. But since the continuity of the cell was lost in the constraint of 20%, other steps of optimization were performed with a constraint greater than this value to calculate the most optimal cell. According to Table 5 and Fig 8, the example of Row No. 4 was chosen as the optimal example of the cubic unit cell due to having the lowest weight.

The results of the optimization for the cylindrical unit cell, with a diameter of 1 m and the height of 1 m, under the same conditions are shown in Table 7 and Fig 10, the stress contours of the optimized structures are shown in Fig 11, and the comparison between the results are presented in Table 8. Similarly, for the cylindrical unit cell, the optimization was done with 4 types of weight constraints. If the remaining weight was less than 20%, the cell was still continuous. Therefore, the optimization was performed with lower percentages to reach the most optimal cell. Finally, the reason for choosing the optimal cell was as follows,

- According to Table 7 and Fig 10, the cell with a weight limit of less than 5% (Row No. 7) did not have continuity. Then, it was not selected.

- According to Table 8, the stress in the cell with a weight limit of less than 10% (Row No. 6) was higher than the stress in the selected cells (HC, TC, and RD). Then, it was not selected again.

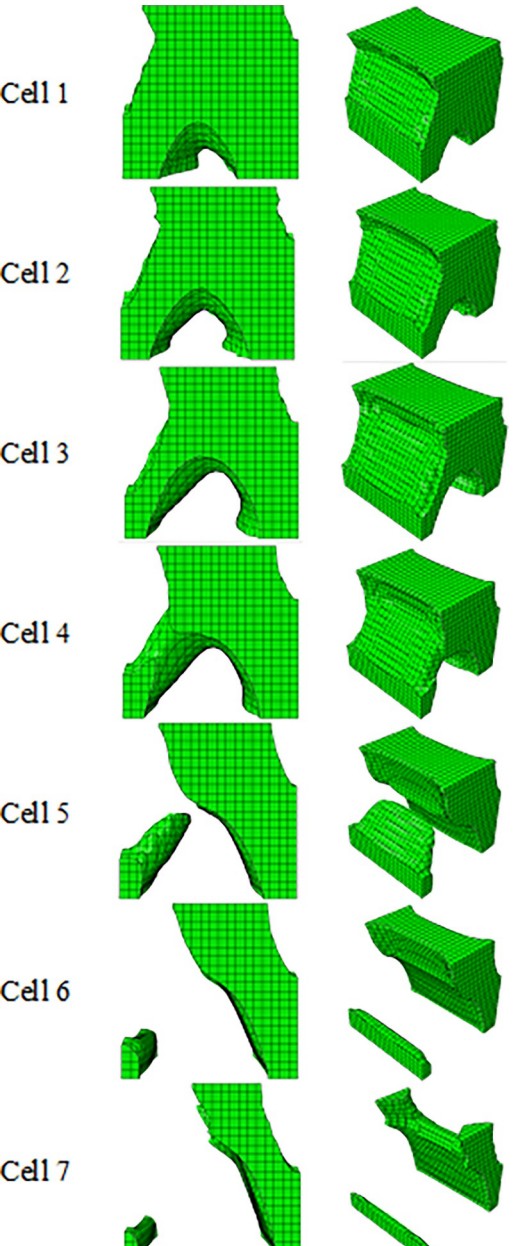

**Fig 8. The results of cubic unit cell optimization for side and isometric views based on the data in Table 5.**

- The cell with a weight limit of less than 15% (Row No. 5) had a very small weight change in addition to a high increase in the stress, compared to the cell with a weight limit of less than 20% (Row No. 4). Then, it was also not selected.

Therefore, the sample of Row No. 4 in Table 7 was selected as the optimal sample.

As a result, in the optimization of the cubic unit cell of the sample with a weight of less than 35% of the initial weight, and the optimization of the cylindrical unit cell of the sample with a weight of less than 20% of the initial weight, it was selected as the most optimal mode. Compared to these two samples, considering that the maximum weight reduction was intended,

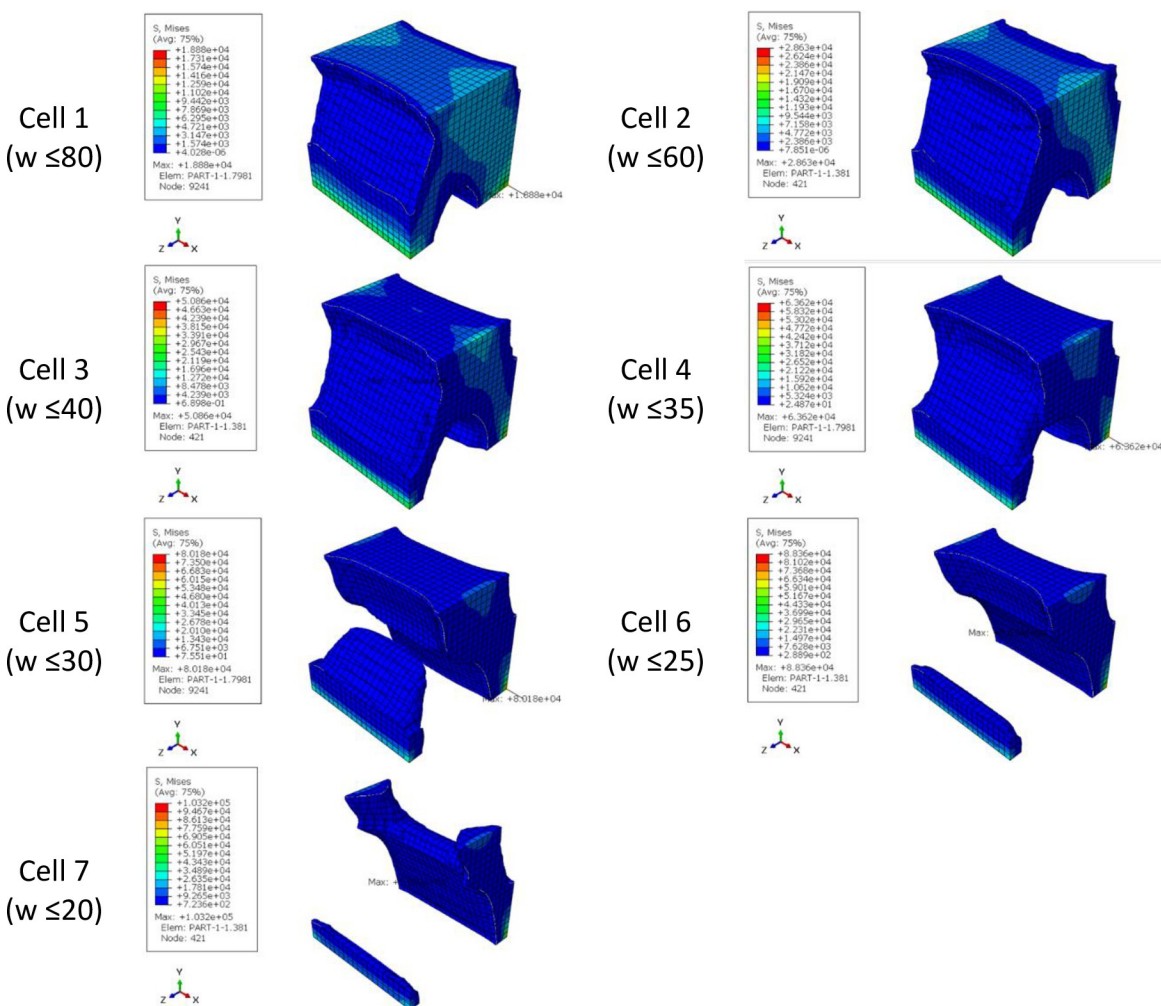

**Fig 9. The von-Mises stress (Pa) contour of the optimized cubic unit cell (The cell number is according to the rows of Table 5 and Fig 8).**

the cylindrical optimized sample was considered the last choice. For better comparison, the results of these two modes are shown in Table 9 and Fig 12.

To fabricate the test specimens, cells were arranged next to and on top of each other. According to the selected cell, there was no possibility of placing the cells on top of each other and producing it. Therefore, in this step, the condition of keeping the initial and final levels constant was used. This issue caused the stress to decrease by 5% and the weight to increase by 2%.

In addition, it was intended to use 3D printers to fabricate the final geometry. As mentioned before, the direction of 3D printing and the overhang control affected the final result. For this reason, for selected cell (Cell 2 in Table 9 and Fig 12) the horizontal print direction suitable for the fatigue test and the vertical print direction suitable for the compressive test was included as a geometric constraint in the optimization. The new results are shown in Table 10 and Fig 13. The stress contour obtained from the resulting geometries is also presented in Fig 14. According to the results, the 3D printing of the model with vertical layering required the use of support structures. In the metamaterial, due to the repetition of cells next to each other, it made it difficult to separate the supporting structures from the main part. Therefore, the

**Table 6. The percentage of changes in stress and weight of cubic unit cell (The cell number is according to the rows of Table 5 and Figs 8 and 9).**

| Cell | The changes in cell stress compared to the stress of cubic unit cell (%) | Changes in the weight of cells compared to the weight of cubic unit cell (%) |
|---|---|---|
| 1 (w ≥80) | 20 | -20.2 |
| 2 (w ≥60) | 86 | -40.9 |
| 3 (w ≥40) | 233 | -60.7 |
| 4 (w ≥35) | 324 | -65.5 |
| 5 (w ≥30) | 433 | -70.4 |
| 6 (w ≥25) | 486 | -75.4 |
| 7 (w ≥20) | 586 | -80.3 |
| HC | 453 | -60.6 |
| TC | 406 | -25.5 |
| RD | 580 | -75.0 |

*A negative value is equivalent to a decrease and a positive value is equivalent to an increase.

model with horizontal layering was chosen. The convergence diagram of the objective function for the selected cell is shown in Fig 15.

To find the smallest 3D printable cell, this cell was produced on different scales. The image of these cells is shown in Fig 16. As shown in this figure, the FFF 3D printer was not capable of producing the smallest scale. This issue was due to the diameter of the nozzle (0.2 mm). If laser-based printers are used, due to the higher accuracy, cells with smaller scales can be produced. Finally, this cell with a length of 8 mm was used in fatigue standard samples with the number of 5 cells and in the compressive standard specimen with the number of 2 cells.

The cut view of the geometry designed for the fatigue test is shown in Fig 17. Similarly, the cut view produced for the compressive test is also presented in Fig 18.

### 3.2) Optimization of fatigue testing cylindrical sample

The results of optimizing the Fatigue Testing Cylindrical Sample (FTCS) with the objective function of minimum strain energy made of PLA material with dimensions of 9x60 mm under

**Table 7. The results of cylindrical unit cell optimization.**

| Cell | Weight Constraint (%) | Iteration* | Max. Stress (Pa) | Final weight (kg) |
|---|---|---|---|---|
| Basic geometry | - | - | $1.8 \times 10^4$ | 973.890 |
| 1 | ≥80 | 26 | $2.1 \times 10^4$ | 777.983 |
| 2 | ≥60 | 31 | $2.6 \times 10^4$ | 583.495 |
| 3 | ≥40 | 35 | $3.7 \times 10^4$ | 389.013 |
| 4 | ≥20 | 17 | $9.5 \times 10^4$ | 189.886 |
| 5 | ≥15 | 16 | $1.3 \times 10^5$ | 142.139 |
| 6 | ≥10 | 15 | $2.2 \times 10^5$ | 94.638 |
| 7 | ≥5 | 23 | $4.8 \times 10^5$ | 48.083 |

*Repeat the solution steps until reaching the final answer.

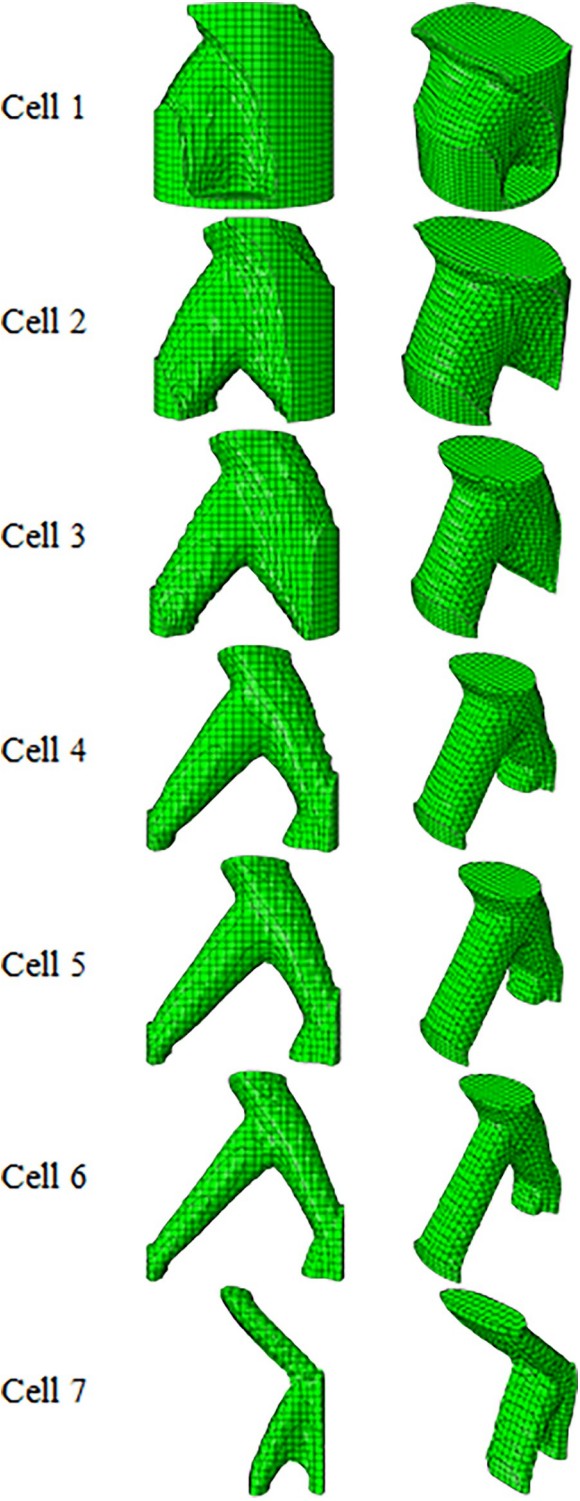

**Fig 10. The results of cylindrical unit cell optimization for side and isometric views based on the data in Table 7.**

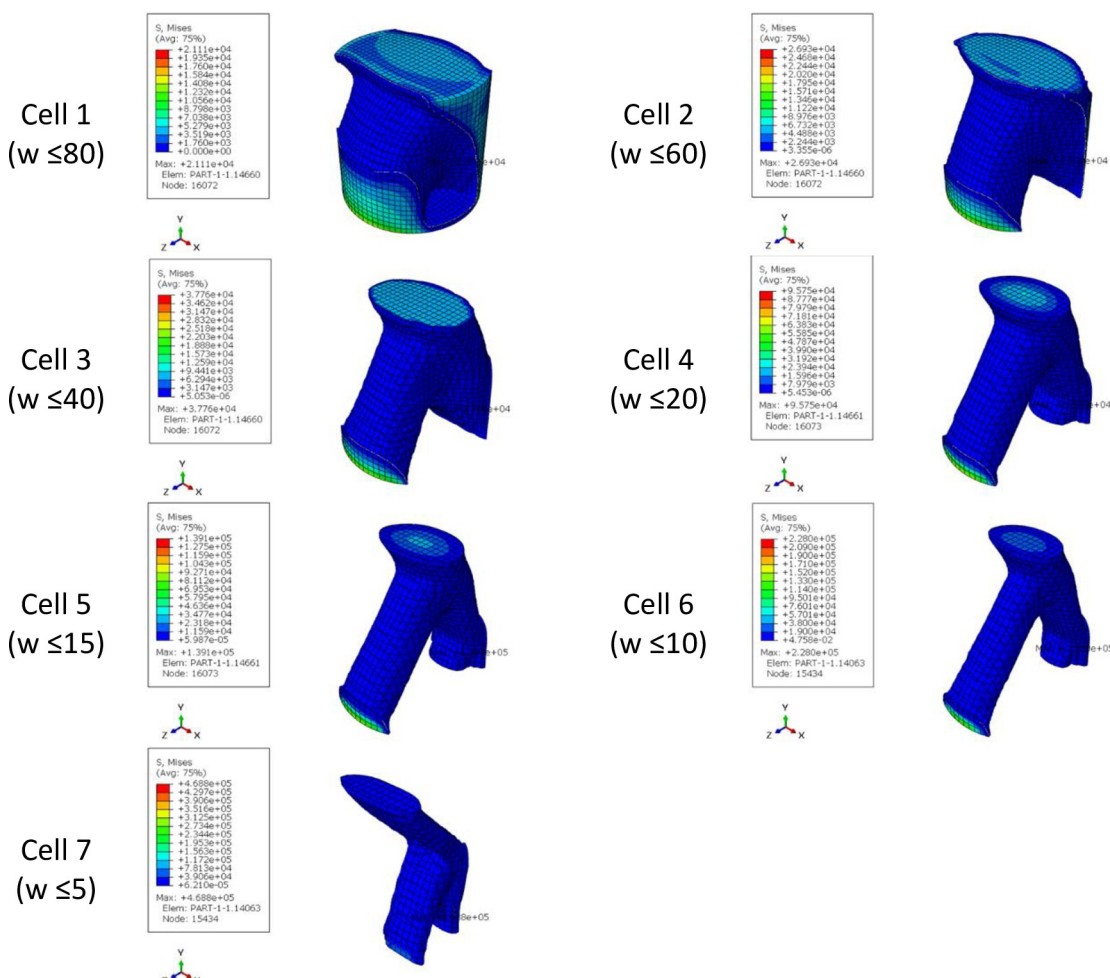

**Fig 11. The von-Mises stress (Pa) contour of optimized cylindrical unit cell (The cell number is according to the rows of Table 7 and Fig 10).**

the concentrated compressive and bending force of 1 kN are in Table 11 and Fig 19 and the stress contour of the optimized structures is presented in Fig 20. Moreover, to make a better comparison between the optimized cells and the initial geometry, the percentage changes in the weight and stress of the cells are presented in Table 12. It should be noted that the result of this optimization was directly placed in the fatigue test machine. The sample was closed by 6 screws at a distance of 1 cm from the beginning and the end part of the sample. Therefore, this part of the specimen was freeze. In fact, the geometric constraint of freezing was considered for this part of the sample. According to the relevant tables in the sample of the fourth row, since the remaining weight tended to zero, the problem had no answer. The sample of the third row was also in the middle area, the thickness of the sample was so thin that it could not be fabricated with the FFF 3D printer. Therefore, the sample of the second row was selected as the optimal sample. It is worth noting that the calculated stress in all cases was higher than the stress of the solid sample and the value of the yield stress of the PLA material. Therefore, failure will occur in the sample. However, considering that the force applied in the simulation process was a coefficient of the real force on the tire (close to the real value), the optimized geometry could be used. Moreover, the amount of force did not affect the optimization of the geometry and only changed the amount of the final stress.

**Table 8. Percentage changes of stress and weight of cylindrical unit cell (The cell number is according to the rows of Table 7 and Figs 10 and 11).**

| Cell | The changes in cell stress compared to the initial stress of the cylindrical unit cell (%) | Changes in the weight of the cells compared to the initial weight of the cylindrical unit cell (%) |
|---|---|---|
| 1 (w ≥80) | 16 | -20.1 |
| 2 (w ≥60) | 44 | -40.0 |
| 3 (w ≥40) | 105 | -60.0 |
| 4 (w ≥20) | 427 | -80.5 |
| 5 (w ≥15) | 672 | -85.4 |
| 6 (w ≥10) | 1166 | -90.2 |
| 7 (w ≥5) | 2500 | -95.0 |
| HC | 361 | -49.8 |
| TC | 322 | -5.1 |
| RD | 466 | -68.1 |

*A negative value is equivalent to a decrease and a positive value is equivalent to an increase.

The overhang control was applied to the selected sample, the results of which are presented in Table 13 and Fig 21. The stress contour of these structures is also shown in Fig 22. According to these tables, the overhang control did not affect the optimization response in the direction of vertical layering. However, changes were made in the direction of horizontal layering and the voids at the beginning and end of the sample were removed. This issue caused the stress in this sample to be lower than the vertical direction. Both of these samples were selected as optimal samples and subjected to fatigue load. The convergence diagram of the objective function for the optimized sample with horizontal layering is presented in Fig 23 and for the sample with vertical layering in Fig 24. The produced samples are also shown in Fig 25.

### 3.3) Fatigue testing results

In this section, two types of samples were produced with the help of 3D printers and subjected to fatigue tests. These two types of samples included optimized FTCS under the title TO and fatigue samples with a cellular structure under the title CS. Each of these samples in different conditions including overhanging directions (2 condition), layer orientation (2 conditions), load level (2 levels), different load point (DLP) (2 conditions) and wall thickness (2 values) ($t$ in Fig 26), were fabricated and tested. The test conditions and information related to each test are shown in Fig 26.

**Table 9. The comparison between optimal cubic and cylindrical geometries.**

| Cell | Weight Constraint (%) | Iteration* | Max. Stress (Pa) | Final weight (kg) |
|---|---|---|---|---|
| 1 | ≥35 | 8 | $6.3 \times 10^4$ | 427.100 |
| 2 | ≥20 | 17 | $9.5 \times 10^4$ | 189.886 |

*Repeat the solution steps until reaching the final answer.

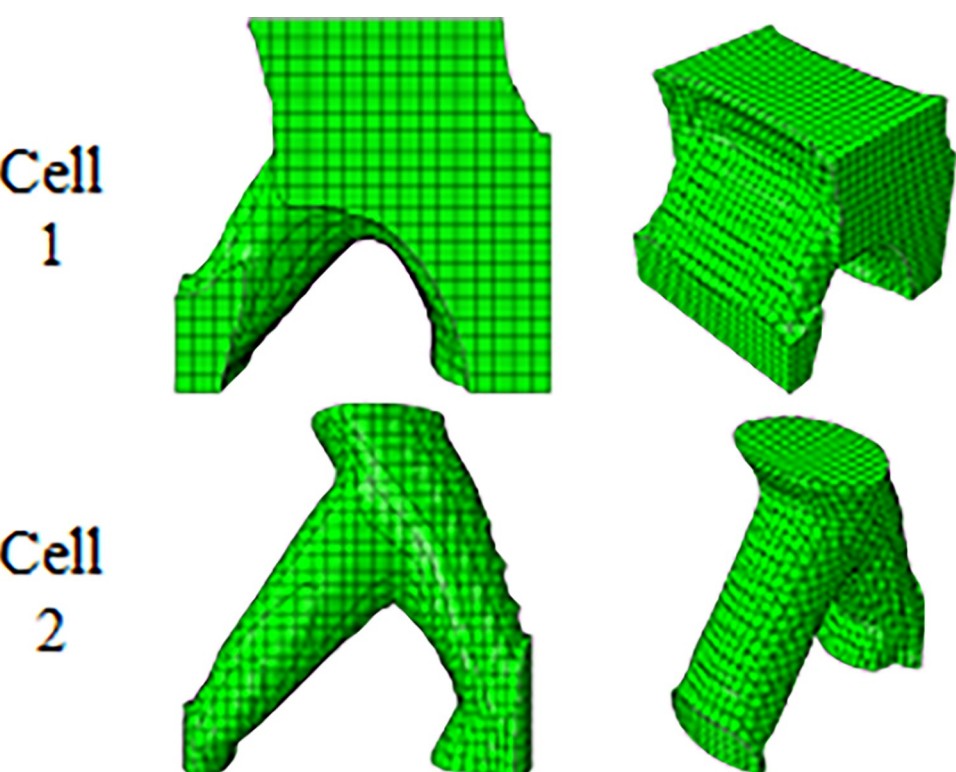

**Fig 12. The comparison between optimal cubic and cylindrical geometries for side and isometric views based on the data in Table 9.**

Mechanical properties of metamaterial are often normalized their weight. In fact, structures that can bear all kinds of mechanical loads with low weight are of special importance. Therefore, the Pareto chart in Fig 27 shows the normalized lifetime of the samples (the ratio of the lifetime of each structure to the maximum lifetime of all samples) according to the difference in weight of each structure compared to the original completely solid structure in a normalized form (the ratio of the difference in weight to maximum weight) is provided. Azadi et al. [77] also considered the weight of each structure to observe the effect of metamaterials on the cyclic bending fatigue test of PLA samples. In fact, in that research, the applied stress level and lifetime of each sample was divided by its weight. However, in the current study, the effect of weight was included in the Pareto chart. In addition, in the present work, the data was used in a normalized form. In the case of research by Azadi et al. [77], the normalization was considered by dividing stress by weight and lifetime by weight.

According to the results in Fig 27, in general, it can be said that the normalized lifetime of TO and CS samples is the same on average and was around 0.7. However, the weight difference

**Table 10. Geometric constraints on the optimization response of cylindrical unit cell.**

| Cell | Layering direction | Iteration* | Max. Stress (Pa) | Final weight (kg) |
|------|--------------------|------------|------------------|-------------------|
| 1 | Horizontal | 47 | $9.0 \times 10^4$ | 194.513 |
| 2 | Vertical | 52 | $9.0 \times 10^4$ | 194.508 |

*Repeat the solution steps until reaching the final answer.

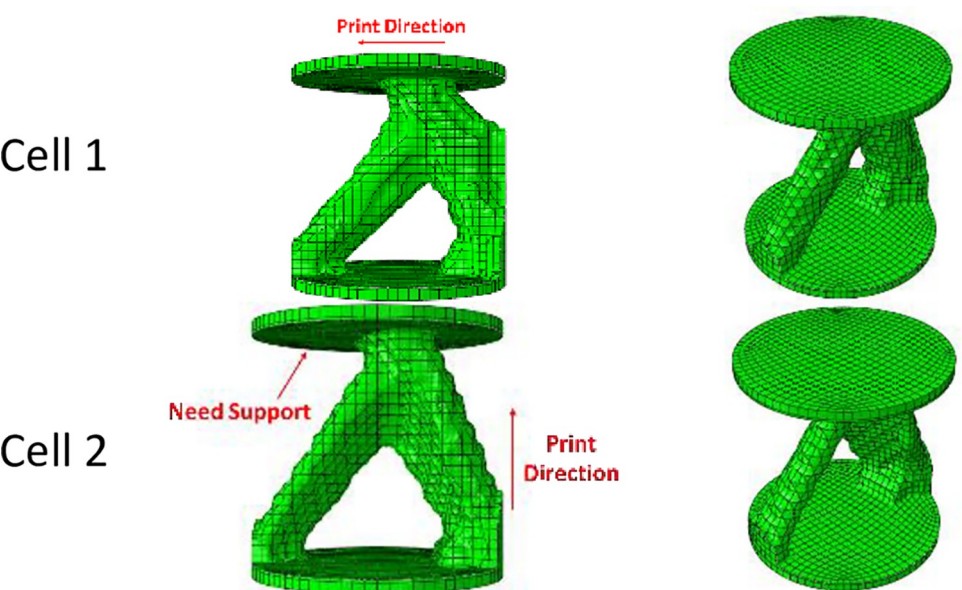

**Fig 13. Geometric constraints on the optimization response of cylindrical unit cell for side and isometric views based on the data in Table 10.**

between these two samples was large. The CS samples had the same lifetime despite their lower weight.

From another point of view, as mentioned earlier, NPTs are lighter than pneumatic tires due to the lack of a tire wall (no need for a tube shape). Therefore, the objective function of the lifetime of the structure may be more important compared to the objective function of weight. In this case, the designer can use CS specimens with a wall thickness of 1 mm. Since these samples had the longest lifetime compared to other samples. It is worth mentioning that although these samples were not lightweight, their average weight was 4.8% less than TO samples and 36% lower than the basic geometry.

It is stated in the definition of metamaterial structures, these structures are created from the repetition and juxtaposition of optimal cells. In this research, it is clear that only the optimization of a structure cannot lead to improved properties. However, these characteristics are obtained from the placement of cells together.

In addition to weight, it was observed in CS cellular structures that the lifetime of the structure was longer if a higher load was applied. The sample CS: Z-H-160(gr) had a longer lifetime

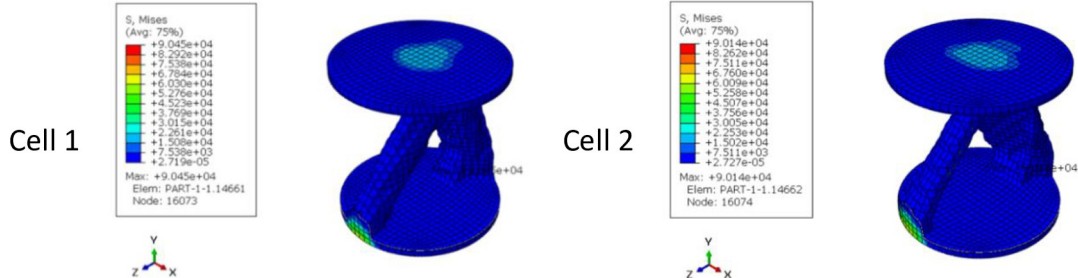

**Fig 14. The von-Mises stress (Pa) contour of optimized cylindrical unit cell (The sample number is according to the rows of Table 10 and Fig 13).**

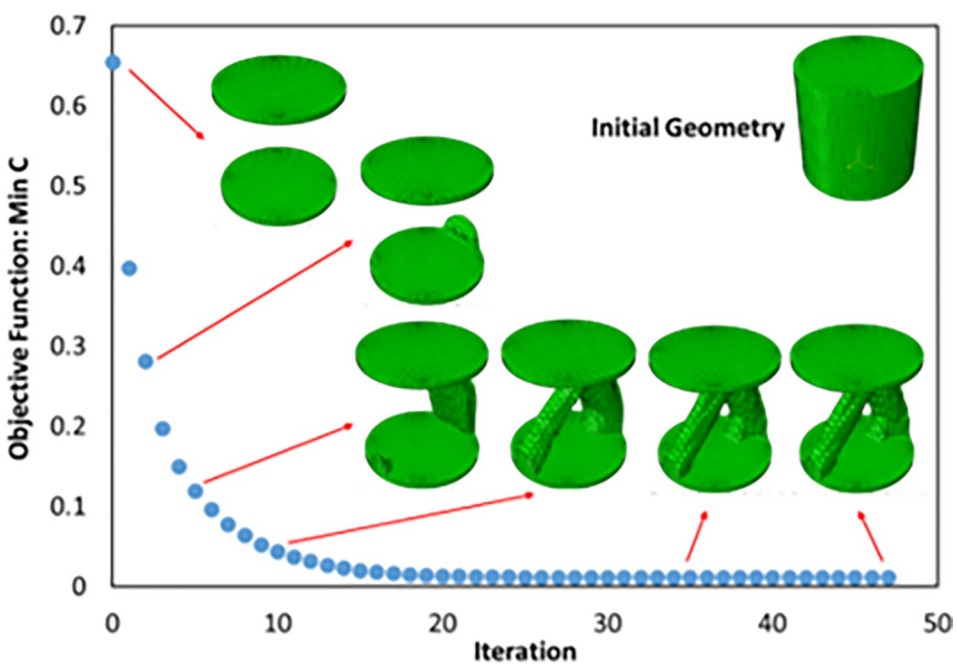

**Fig 15. Objective function convergence diagram for cylindrical cell.**

compared to the sample CS: Z-H-60(gr), which was against the natural behavior of the material. To ensure this issue, the sample CS: Z-H-60(gr) was tested twice. Similarly, the sample CS: Z-H-160(gr)-DLP, also had a longer lifetime.

This observation may occur during the lifetime of the samples for two reasons. The first reason can be due to the dispersion range of metamaterial life data in the fatigue phenomenon. To prove this, it is necessary to enhance the number of fatigue tests. In this case, the reliability of fatigue lifetime data can be calculated. This can be a suggestion for future work. Another reason could be due to the abnormal behavior of the material with negative stiffness (NS) properties, which was justified by Chen et al [96]. For NS metamaterials, increasing

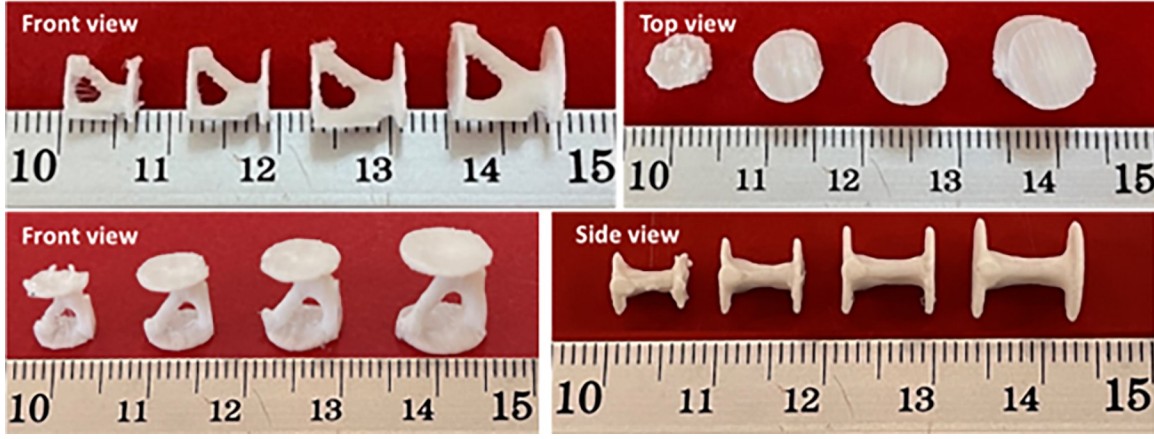

**Fig 16. The scaled cells after manufacturing with a 3D PLA printer.**

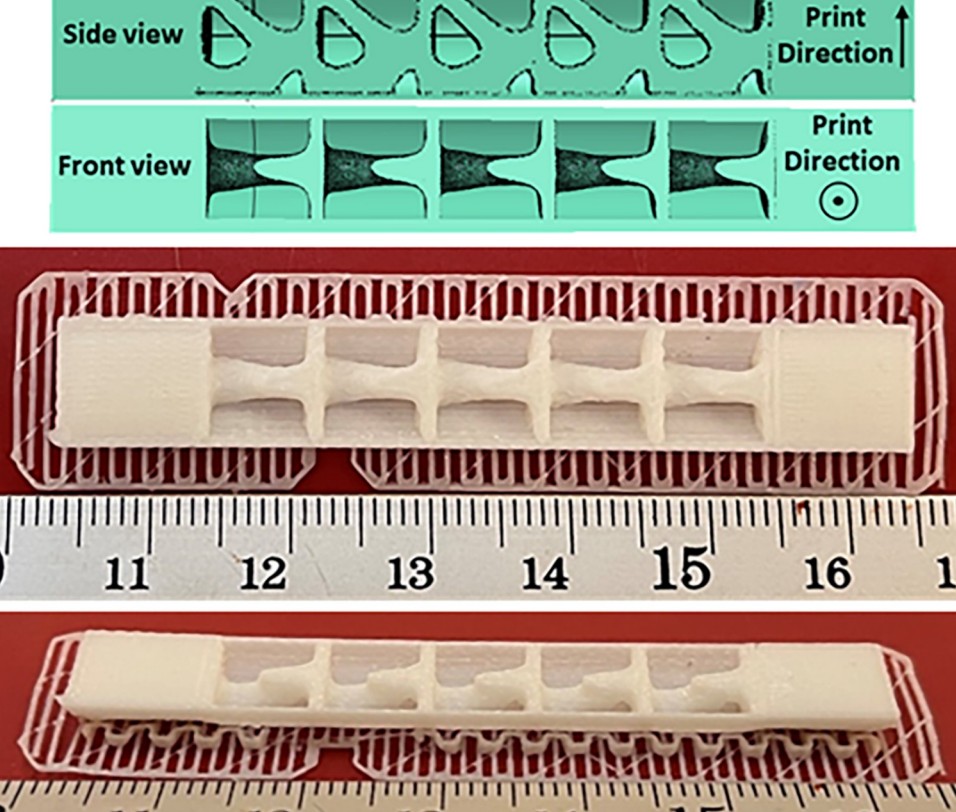

**Fig 17. The sample designed and produced for fatigue test.**

deformation leads to load loss. To accurately prove this claim, it is necessary to calculate the stress distribution and strain energy absorption, which can be done in future works.

Similar tests were repeated for TO: Z-H-60(gr) and TO: Z-H-160(gr) samples. However, since the definition of metamaterial is not true in these structures, behavior similar to cellular structures was not observed. It is worth mentioning that the sample TO: Z-H-60(gr) was also tested twice.

This behavior was not observed in CS specimens with a cell wall thickness of 1 mm. One reason can be caused by the change of cell geometry and the other reason can be caused by the dispersion range of metamaterial life data in the fatigue phenomenon.

It was already stated that horizontal layering had a better performance than vertical layering in the fatigue test. The same result was obtained by comparing the results of TO: Z-H-160(gr) and TO: Y-V-160(gr) samples.

### 3.4) Compressive testing results

The compressive test was performed for two types of solid and cellular samples. For each of these samples, the compressive test was repeated in 3 stages. In addition, a sample with cellular structure and different load point (DLP) was tested. In general, 7 samples were produced and tested, the information of which is presented in Fig 28. In Fig 29(A), the stress-strain diagram resulting from the compressive test of these samples is presented. In this figure, the outer

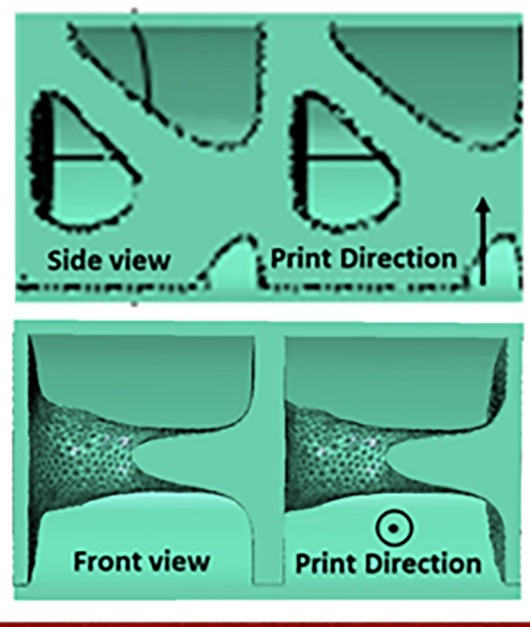

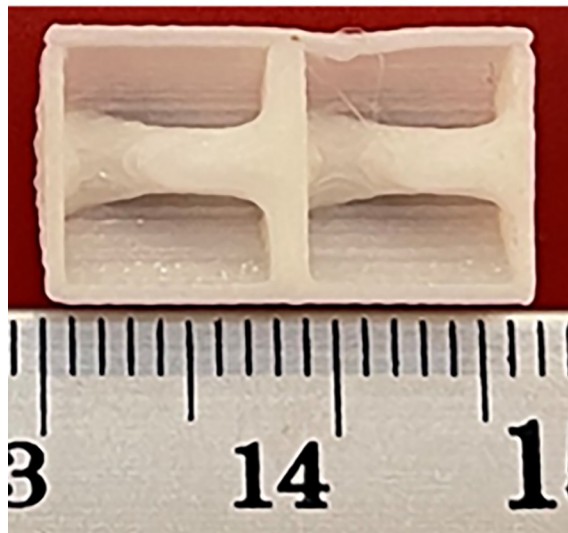

**Fig 18. The sample designed and produced for compressive test.**

**Table 11. The results of the FTCS optimization.**

| Sample | Weight Constraint (%) | Iteration* | Max. Stress (Pa) | Final weight (kg) |
|---|---|---|---|---|
| Basic geometry | - | - | $9.06\times10^8$ | 0.004 |
| 1 | $\geq80$ | 36 | $9.06\times10^8$ | 0.003 |
| 2 | $\geq60$ | 37 | $9.89\times10^8$ | 0.002 |
| 3 | $\geq40$ | 46 | $6.11\times10^9$ | 0.001 |
| 4 | $\geq20$ | - | - | - |

*Repeat the solution steps until reaching the final answer.

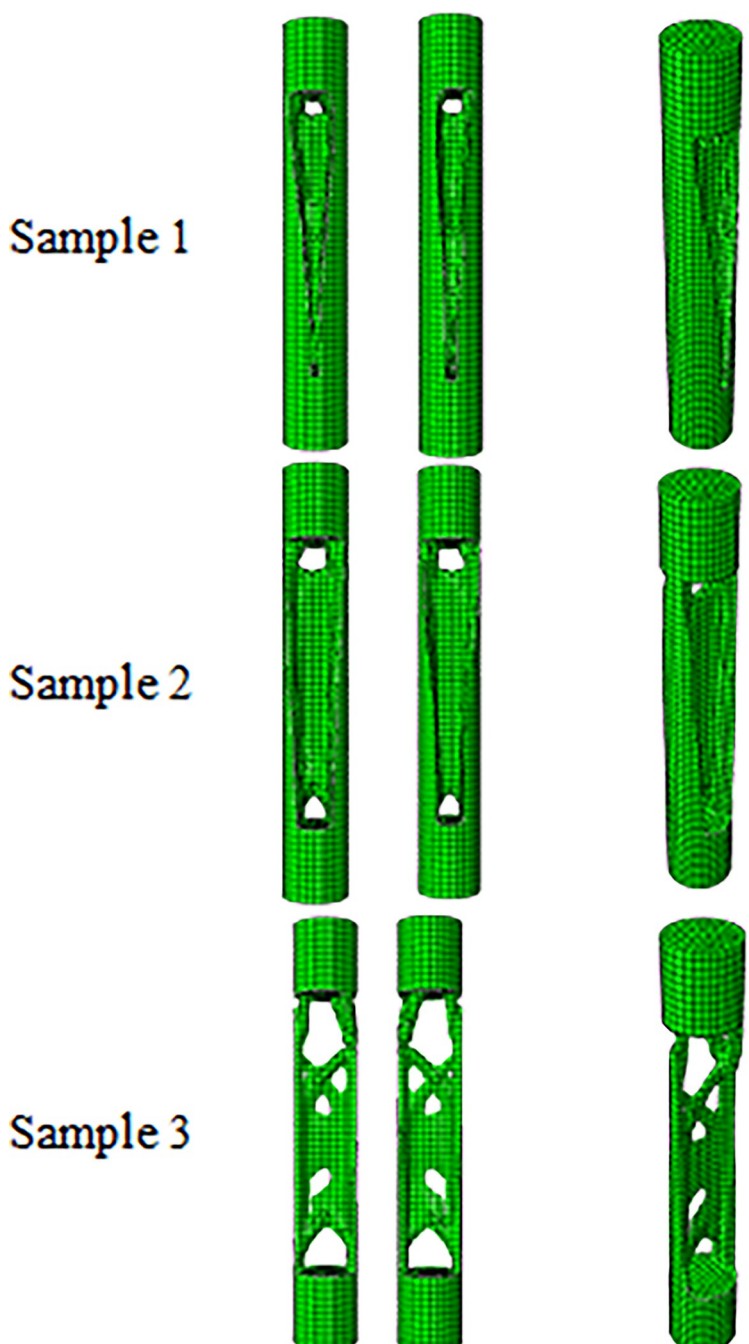

**Fig 19. The results of the FTCS optimization for side and isometric views based on the data in Table 11 (no existed geometry for Sample 4).**

dimensions were used to calculate the stress. Wickeler and Naguib [69] also used the stress-strain diagram in metamaterial samples in the compressive test. However, since the cross-sectional area in cellular structures was variable and cannot be calculated correctly, according to the work of Namvar et al. [68] the displacement force diagram was shown for these samples in part (b) of the same figure (Fig 29(B)). Similar to what was said, the mechanical properties are compared to the weight of the structure. Therefore, the amount of force relative to the weight

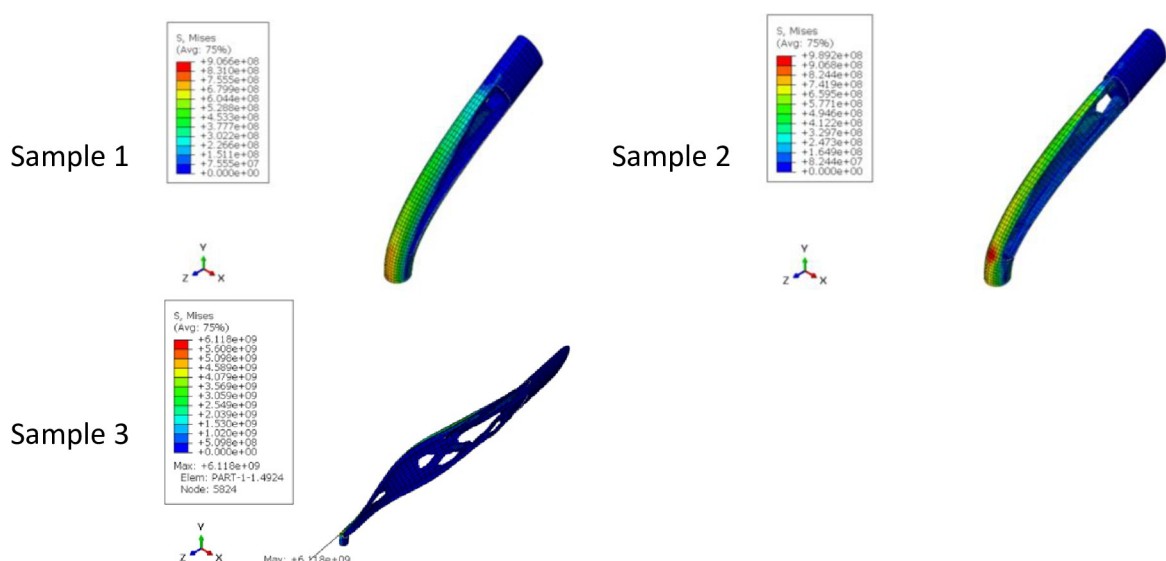

**Fig 20. The von-Mises stress (Pa) contour of optimized FTCS (The sample number is according to the rows of Table 11 and Fig 19. Moreover, no answer was obtained for Sample 4).**

of each structure can also be seen in Fig 29(C). Kshad et al. [97] compared the compression properties of metamaterial structures in relation to their weight.

As it can be seen, although the cellular structure did not lead to an increase in the yield stress compared to the original geometry, but considering the weight, the difference between the maximum force tolerated in the cellular and solid structures was reduced. In fact, according to the obtained results, a 59% significant reduction in weight resulted in a 47% reduction in yield stress relative to weight.

Fig 30 also shows the maximum normalized force (the maximum force in the linear region of each structure compared to the maximum force of all samples) compared to the normalized weight difference (the difference in the weight of each structure and the solid structure compared to the maximum weight). As mentioned in Section 3.3, in NPTs, it is important to reduce weight and from another point of view increase lifetime. Therefore, according to Fig 30, if weight reduction was more important for the designer, cellular structures can be a better choice.

### 3.5) Examining failures and fractures

After carrying out the tensile test for standard samples made of PLA to calculate the properties of the materials, the fracture surface of these samples was investigated. According to this figure,

**Table 12. The percentage changes of stress and geometry weight of FTCS (The cell number is according to the rows of Table 11 and Figs 19 and 20).**

| Sample | Changes in cell stress compared to the initial stress of FTCS (%) | Changes in the weight of the cells compared to the initial weight of the FTCS (%) |
|--------|------------------------------------------------------------------|-----------------------------------------------------------------------------------|
| 1 | 0.000 | -36.5 |
| 2 | 9 | -57.7 |
| 3 | 574 | -78.8 |
| 4 | No answer | No answer |

*A negative value is equivalent to a decrease and a positive value is equivalent to an increase.

Table 13. The effect of overhang condition on the optimization response of FTCS.

| Sample | Weight Constraint (%) | Iteration* | Max. Stress (Pa) | Final weight (kg) |
|---|---|---|---|---|
| 1 | Horizontal | 40 | $1.072 \times 10^9$ | 0.002 |
| 2 | Vertical | 34 | $1.076 \times 10^9$ | 0.002 |

*Repeat the solution steps until reaching the final answer.

it was clear that the PLA samples were broken in plane. Banjanin et al. [98] also observed similar results in the failure of standard tensile test samples made of PLA material and made by the FFF method.

In Fig 31, the holes were observed for PLA samples. Defects in printed polymers are created for two reasons. The first reason is due to the manufacturing method. In fact, during 3D printing, the layers are not properly connected to each other. In other references, the weak

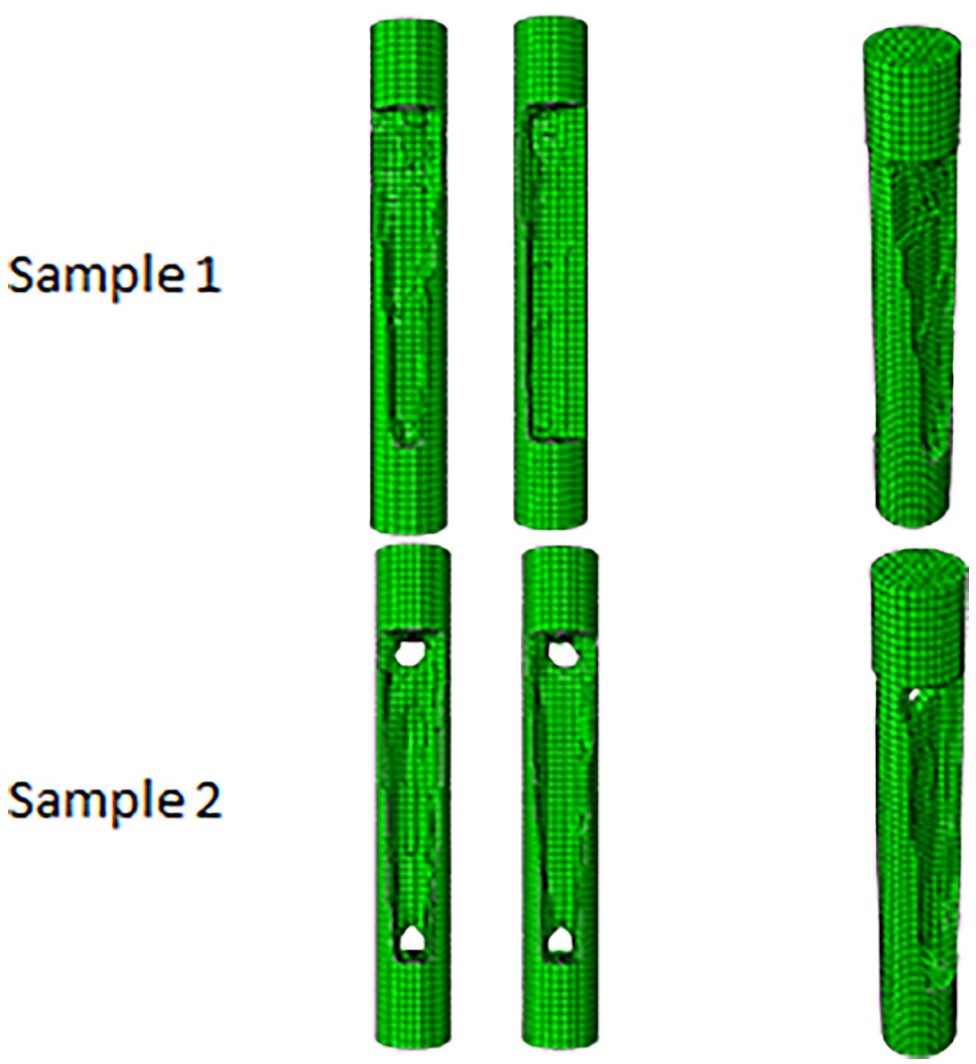

Fig 21. The effect of overhang condition on the optimization response of FTCS for side and isometric views based on the data in Table 13.

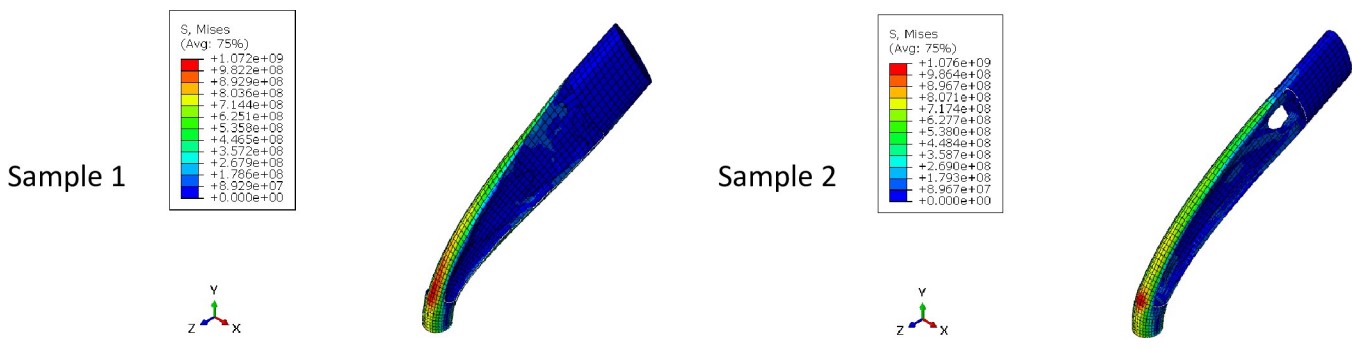

**Fig 22. The von-Mises stress (Pa) contour of optimized FTCS (The sample number is according to the rows of Table 13 and Fig 21).**

connection between layers in the FFF technique was observed [99]. These types of defects can cause the stress concentration and as a result, crack initiation.

Another type of defect was also seen on the failure surface. These defects are not connected in a crack. They are formed during the production of the part due to the high temperature applied to the filament and the creation of gases [75,76]. The first defect had an irregular shape and most importantly slit-shape. Whereas the second defect size was typically small and had spherical shapes.

Moreover, in Fig 31, the fracture surface can be seen in the form of sheets. Similar observations were also reported by Pinto et al. [100] for PLA composite.

As a result of compression, the samples did not fail. Various behaviors such as cracking, buckling, symmetrical bending, and lateral shrinkage of the sample are the effects of compressed samples [100]. Therefore, in this section, damage effects in the samples after the compressive test was investigated. The image of the fracture behavior in compressive testing is

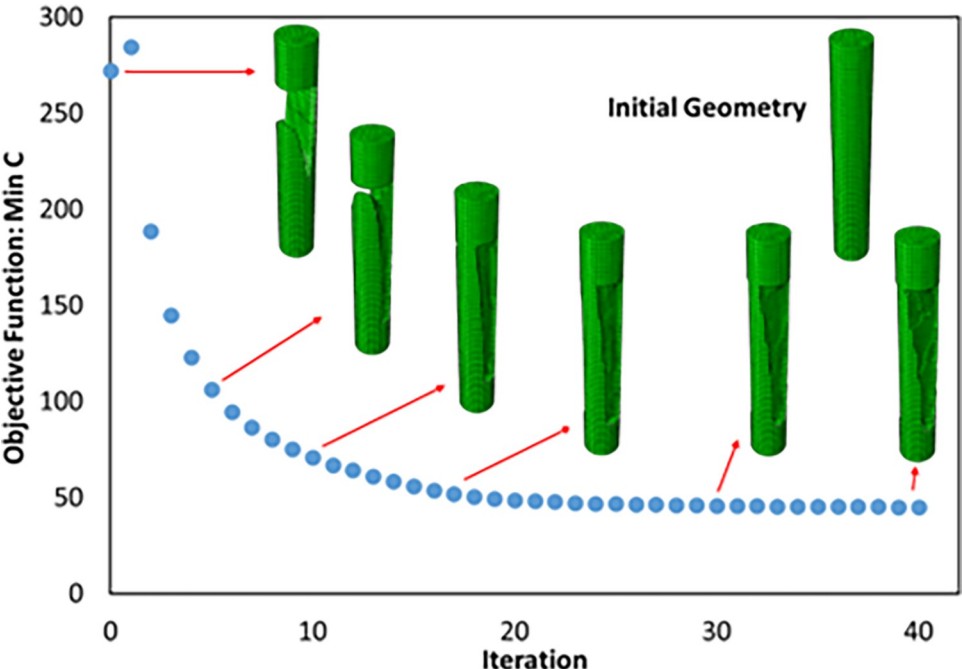

**Fig 23. Objective function convergence diagram for optimized FTCS with horizontal layering.**

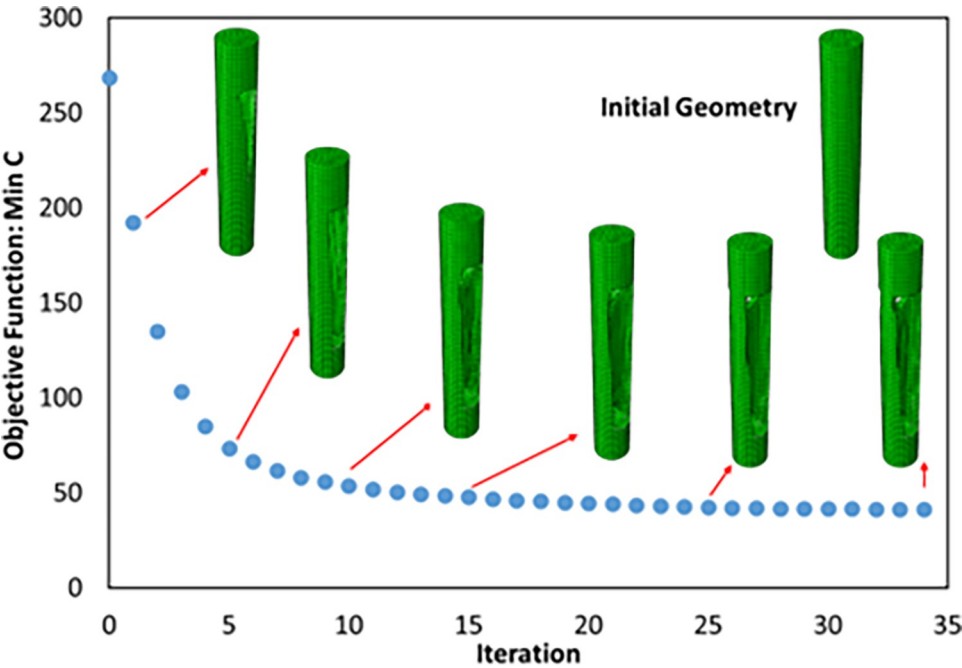

**Fig 24. Objective function convergence diagram for optimized FTCS with vertical layering.**

shown in Fig 32, for both solid and cellular structures. According to these figures, the stress vector applied to both samples by the compressive load was created between the layers. This issue caused the separation of layers. This behavior was also observed by Lim et al. [101].

Dou et al. [102] reached a similar result after horizontal printing of compressive standard PLA samples. They believed that the propagation of cracks caused by compression in the direction of the layers during loading was the cause of failure. They introduced the main cause of the separation of layers due to 3D printing defects and the creation of holes between the layers. Barkhad et al. [103] also fabricated samples of PLA material but printed in the vertical direction and performed the compressive test for these samples. In these samples, the separation between the layers occurred. However, since the direction of the 3D printing was different, the

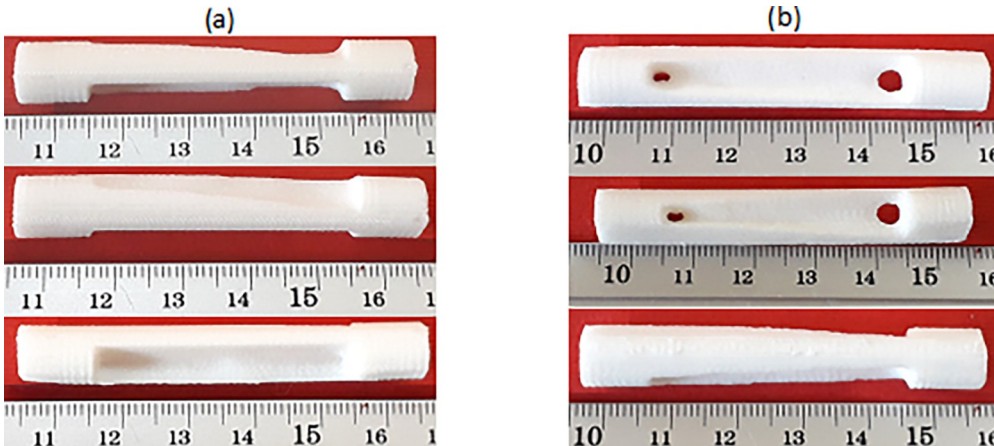

**Fig 25. The optimized fatigue example: (a) vertical layering and (b) horizontal layering.**

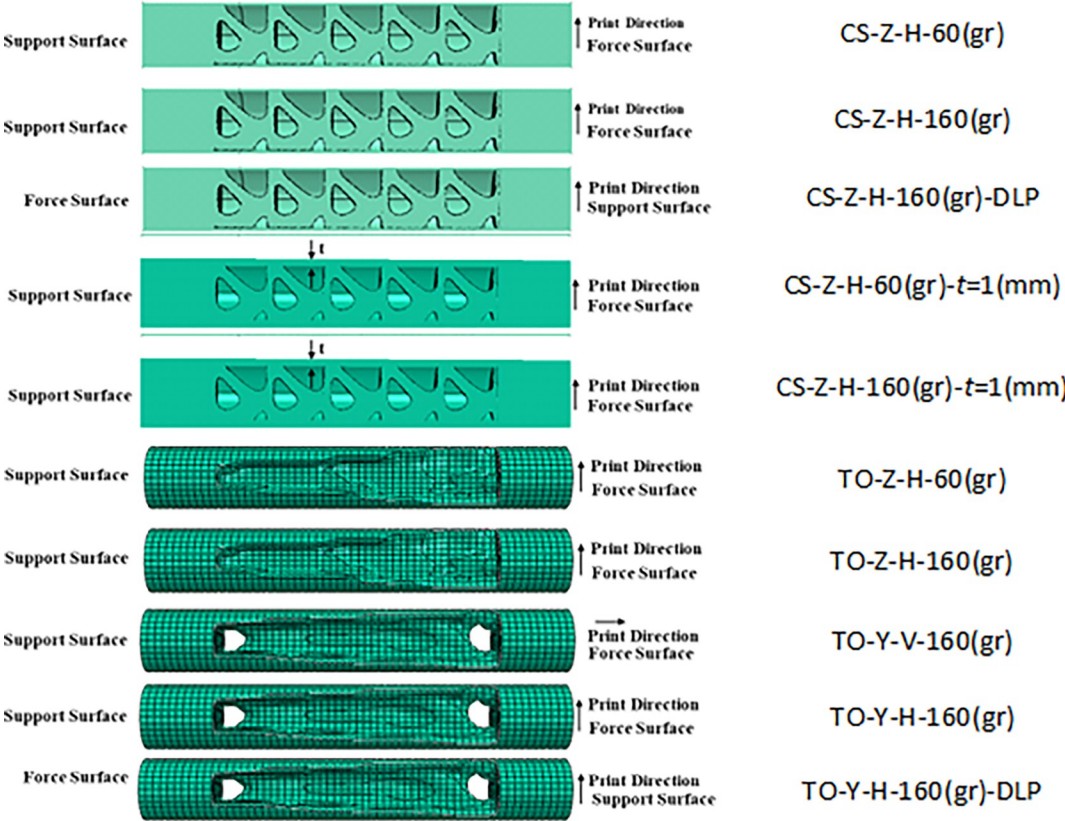

**Fig 26. Different conditions on fatigue testing with the sample name.**

separation of the layers led to the sliding of the layers on each other. Similar behavior was observed for 3D printed samples in the vertical direction by Beniak et al. [104].

The image of the NPT with the optimized cell for the future research is shown in S6 File. Moreover, in further investigations, the following issues will be considered and studied,

- The mechanical properties of this optimized tire including the compressive behavior and the fatigue performance,

- Different arrangements between the unit cell under compressive loads,

- The behavior of hyper-elastic materials such as TPU (as rubbery structures) in the optimization process, besides considering the damping behavior.

- Evaluating the effect of mean stress on the structure performance during fatigue testing.

## 4) Conclusions

In order to design the metamaterial cell, used in non-pneumatic tires (NPTs), three types of geometries, including a cube with unit dimensions, a cylinder with unit dimensions, and a FTCS with two compressive and bending forces, were optimized. Then, two types of optimized structures including CS and TO were obtained. The results of compressive and fatigue testing for these two types of structures including the following remarks,

- According to the capability of FFF 3D printing, the smallest HC unit cell was 4.2 mm for the length.

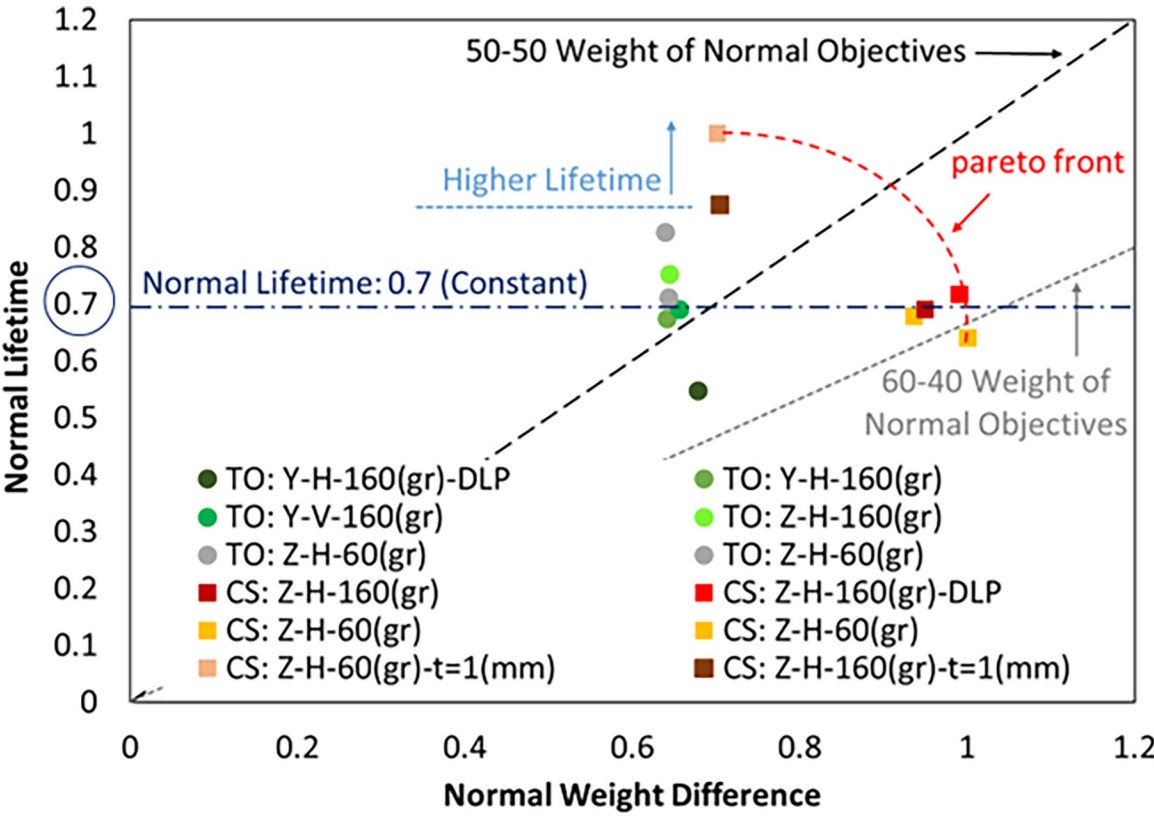

**Fig 27. The Pareto chart for the specimens under fatigue testing.**

- In the optimization of cubic unit cell, the sample with the remaining weight limit of less than 35% was selected as the optimal sample. However, in the optimization of the cylindrical unit cell, the sample with a weight constraint of 20% was the most optimal state.

- Based on the obtained results, in the optimization for the FTCS, the response with the weight limit of 60% was selected as the optimal sample due to the continuous structure and considering the lowest weight.

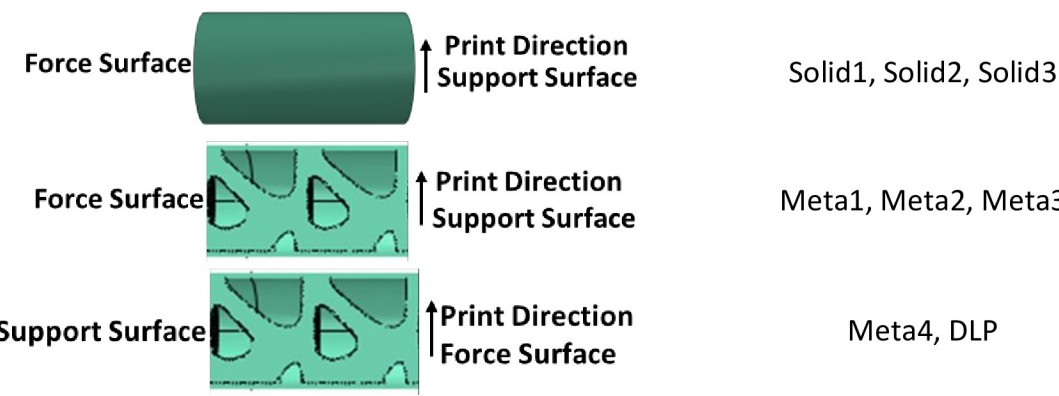

**Fig 28. Different conditions on compressive testing with the sample name.**

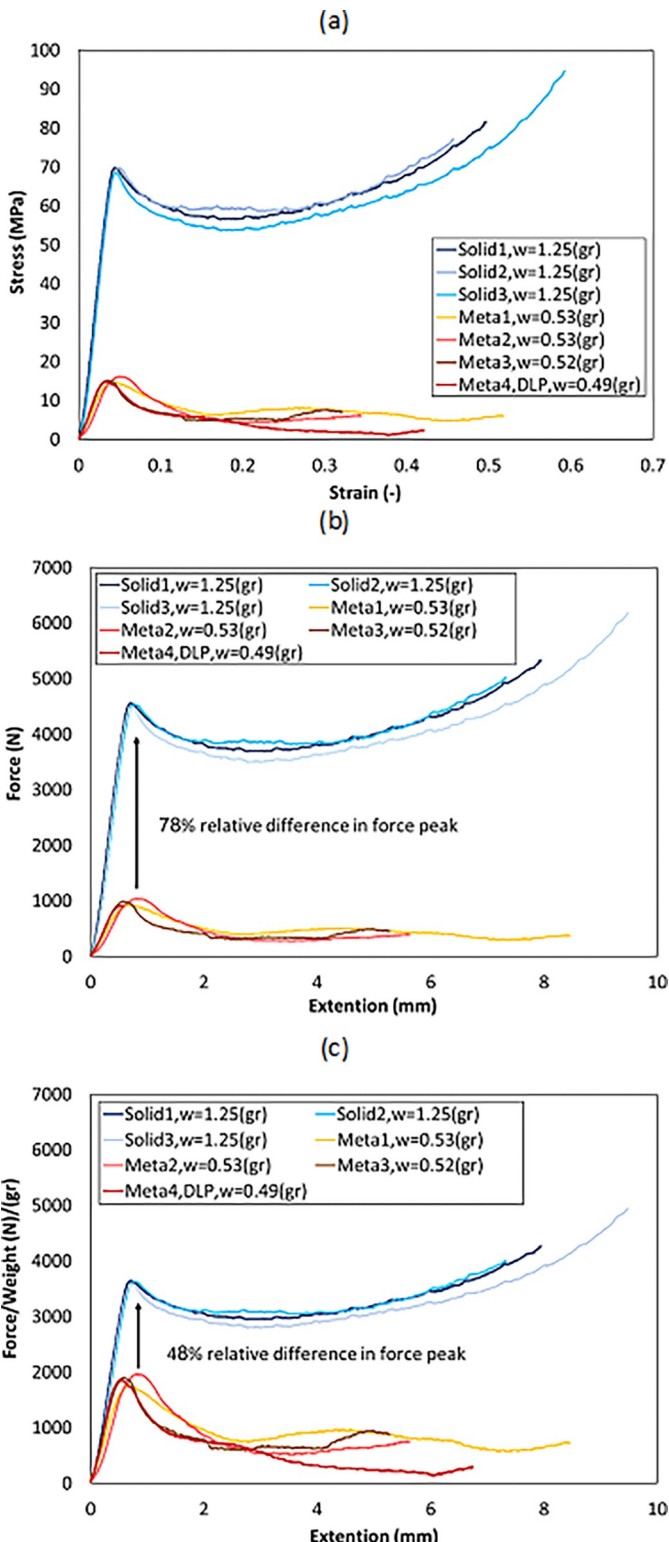

**Fig 29.** Compressive test results for PLA samples: (a) stress-strain diagram, (b) force-displacement diagram, and (c) force-to-weight-displacement diagram.

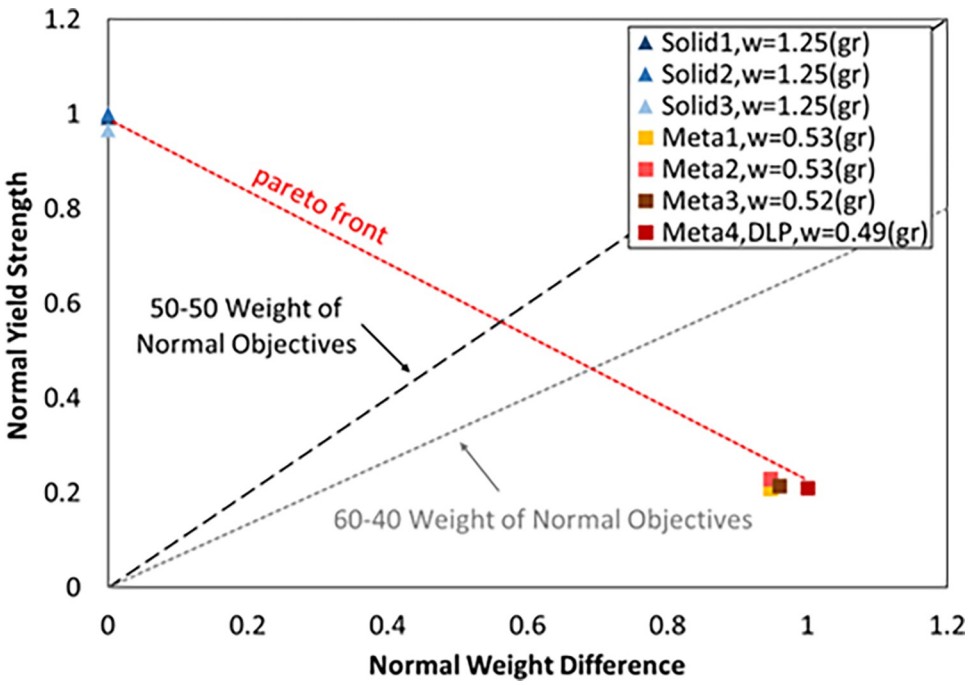

**Fig 30. Normalized yield stress graph compared to the normalized weight difference.**

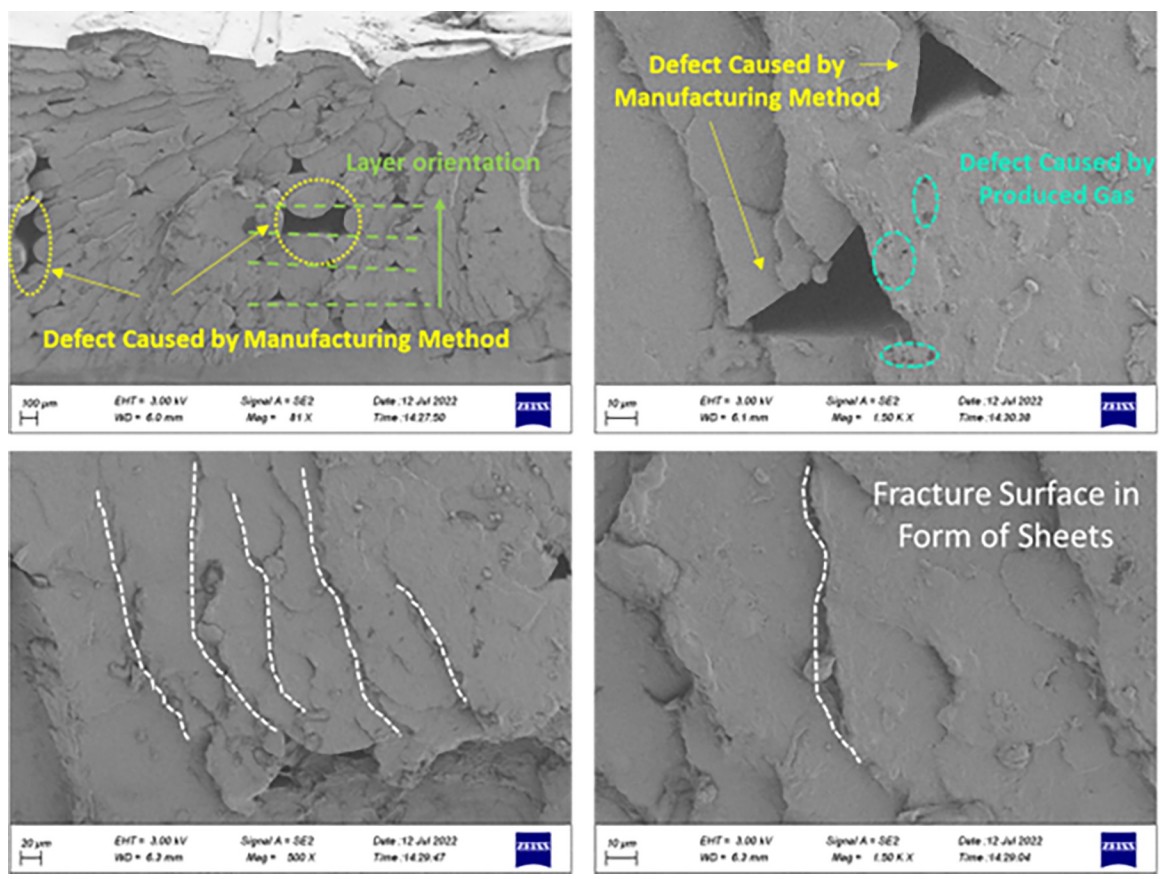

**Fig 31. FE-SEM images for the fractured surfaces of PLA sample under tensile loading.**

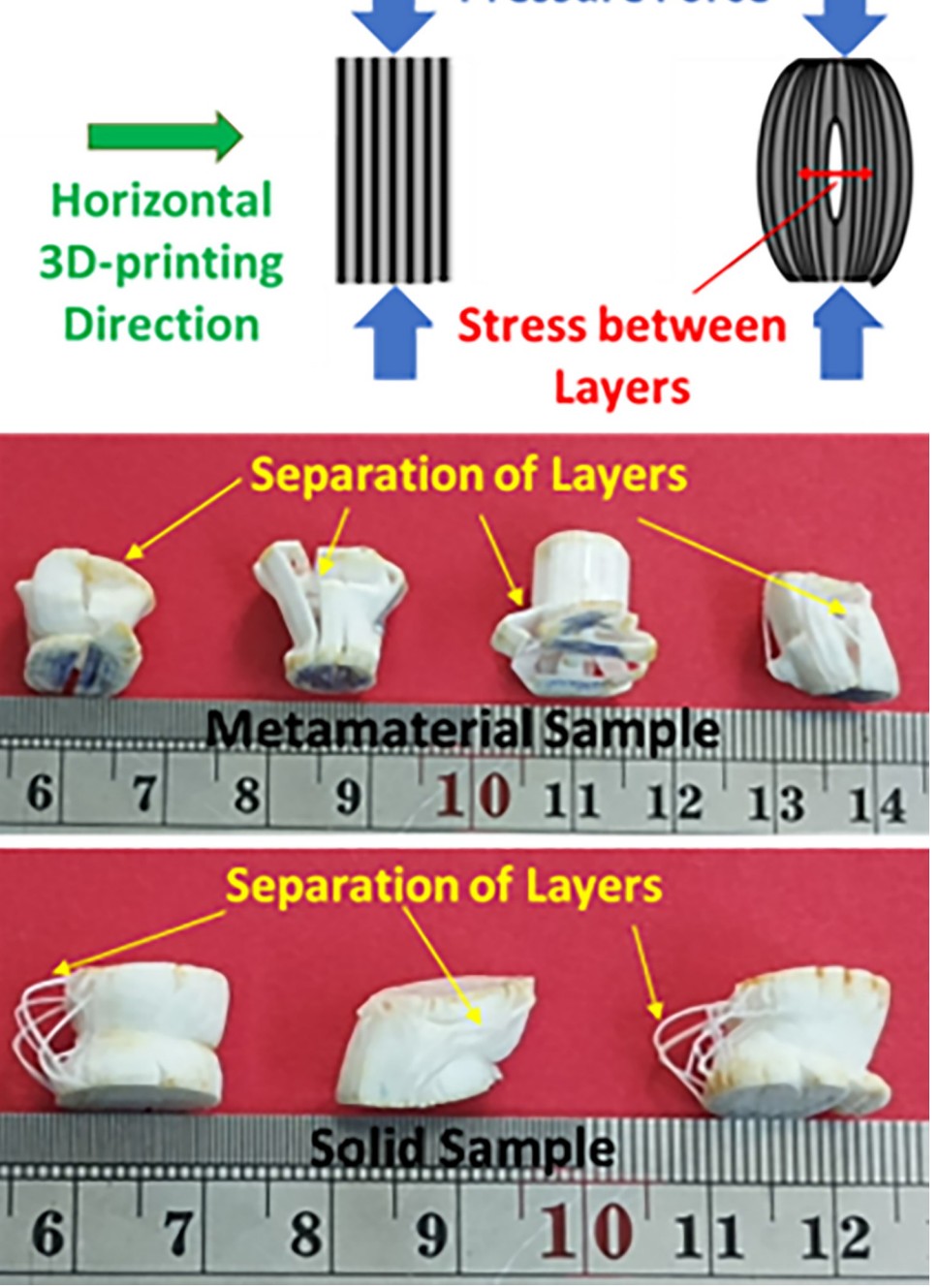

**Fig 32. The fracture behavior in compressive tests, damaged solid and cellular structures.**

- Cellular structures (CS) with 15% of lower weight than the optimized samples had the same fatigue lifetime (0.7 of the normal lifetime). These samples were designed for the application in car tires. Manufacturers are always looking for lower weight in the car. Since the lighter car brings benefits, such as reducing the fuel consumption and emissions. Therefore, if the designers are seeking for the lightweight design of NPTs, they could use this cell type.

- Specimens with a thickness of 1 mm had 38% of higher lifetime and 14% of higher weight than those structures with 0.5 mm of the thickness.

- The average weight of specimens with a thickness of 1 mm was 4.8% less than TO samples and 36% less than solid samples.

- The holes caused by the production method caused a weak connection between the layers and their separation in the tensile test.

- To achieve properties contrary to nature, only optimizing a structure is not enough. Rather, several cells must be repeated.

## Supporting information

**S1 Graphical abstract.**
(DOCX)

**S1 File. A literature review on NPTs.**
(DOCX)

**S2 File. A benchmark on cells and their properties.**
(DOCX)

**S3 File. Tensile testing.**
(DOCX)

**S4 File. Topology optimization.**
(DOCX)

**S5 File. Devices.**
(DOCX)

**S6 File. Arrangement of cells in NPTs.**
(DOCX)

## Author Contributions

**Conceptualization:** Mohammad Azadi.

**Data curation:** Shokouh Dezianian.

**Formal analysis:** Shokouh Dezianian.

**Funding acquisition:** Mohammad Azadi.

**Investigation:** Shokouh Dezianian, Mohammad Azadi, Seyed Mohammad Javad Razavi.

**Methodology:** Mohammad Azadi.

**Project administration:** Mohammad Azadi.

**Resources:** Shokouh Dezianian, Mohammad Azadi.

**Software:** Shokouh Dezianian.

**Supervision:** Mohammad Azadi, Seyed Mohammad Javad Razavi.

**Validation:** Seyed Mohammad Javad Razavi.

**Visualization:** Mohammad Azadi.

**Writing – original draft:** Shokouh Dezianian.

**Writing – review & editing:** Mohammad Azadi, Seyed Mohammad Javad Razavi.

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
