## [Decision Letter · Decision Letter 0]

4 Jun 2023

PONE-D-23-07715Topology Optimization on Metamaterial Cells for Replacement Possibility in Non-Pneumatic Tire and the Capability of 3D-printing with Polylactic Acid FilamentsPLOS ONE

Dear Dr. Azadi,

Thank you for submitting your manuscript to PLOS ONE. After careful consideration, we feel that it has merit but does not fully meet PLOS ONE’s publication criteria as it currently stands. Therefore, we invite you to submit a revised version of the manuscript that addresses the points raised during the review process.

Please, address all the comments made by the reviewers.

We look forward to receiving your revised manuscript.

Kind regards,

Antonio Riveiro Rodríguez, PhD

Academic Editor

PLOS ONE

“This work is funded by Iran National Science Foundation (INSF) under project No. 4002601.”

5. Please ensure that you refer to Figure 26 in your text as, if accepted, production will need this reference to link the reader to the figure.

Reviewers' comments:

Reviewer's Responses to Questions

**Comments to the Author**

1. Is the manuscript technically sound, and do the data support the conclusions?

Reviewer #1: Yes

Reviewer #2: Yes

2. Has the statistical analysis been performed appropriately and rigorously? 

Reviewer #1: N/A

Reviewer #2: N/A

3. Have the authors made all data underlying the findings in their manuscript fully available?

Reviewer #1: Yes

Reviewer #2: Yes

4. Is the manuscript presented in an intelligible fashion and written in standard English?

Reviewer #1: No

Reviewer #2: No

5. Review Comments to the Author

Reviewer #1: The paper presents an interesting analysis on the design of structures for the use in non-pneumatic tires.

However, there are comments and questions that need to be addressed.

1. P.2, l.2 …” Poisson ratio close to zero [1].”- besides reference 1, the authors should also cite the book https://doi.org/10.1017/CBO9781139878326, Lorna J. Gibson and Michael F. Ashby

2. P.2, which are extensively researched and presented in Appendix A. Why did the authors chose to separate this review from the introduction? The text from Appendix A could be compressed and inserted here. In fact, all the introduction should be shortened.

3. P.3“Mazur [9] has compared a non-pneumatic tire with pneumatic tires. It has been concluded that the rolling resistance force of non-pneumatic tires is different from pneumatic tires” how different? Smaller or larger?

4. P.3 “Two types of hexagonal honeycomb are designed: same load carrying capacity and same cell wall thickness.” Please explain the difference between the cell structure.

5. P.3, “Finding a cell with a resistance to compressive forces and bending cyclic loads (fatigue).” Compared to what?

6. P.4, 2.1) Patterning of cells: please add that the cells of Appendix A are used in several applications.

7. P4. “Table 1 in appendix B”, it should be Table B1.

8. Replace all over the text FDM by FFF (fused filament fabrication), as FDM is a trade mark.

9. P.7 “The main material which used is poly lactic acid (PLA).”, please explain the use of a rigid material and not the use of for example a rubber?

10. P.7” The main material which used is poly lactic acid (PLA).” There is a missing word. There are some typos in the text. Please correct it.

11. Appendix C also should be inserted in the text of section 2.3

12. “According to Table 7, the example of row 4 is chosen as the optimal example of the cubic unit cell due to having the lowest weight.” Please explain why it is not row 7.

13. Figure 13 – could the authors discuss the defects obtained in the smaller specimen?

14. P. 22, explain what do you mean by “As mentioned in the third chapter…? What chapter or what section?

15. P.31 “The image of compressed solid samples is shown in part a of Figure 24 and the image of compressed samples with cellular structure is shown in part b of the same figure.”, there is no part a and part b in the caption of the figure.

16. Overall, the authors should shorten the text. The work probably comes from a PhD thesis and it is too long for a unique paper.

Reviewer #2: Manuscript Draft: Topology Optimization on Metamaterial Cells for Replacement Possibility in Non-Pneumatic Tire and the Capability of 3D-printing with Polylactic Acid Filaments

PONE-D-23-07715

General Comments:

- Major English revision is needed. Also, some parts of the document include repetitions which could be consolidated to obtain a more concise and clearer document. Some of the analyses of the results are too extensive and confusing due to the high number of different nomenclature used.

- Can the authors comment on the adequacy of PLA material in the production of car’s tires?

- The authors mention very often that the tire is subject to bending and compressive loads, but there is a chapter explaining the tensile tests performed to obtain mechanical properties of the material used to be provided to the optimization software. Could the authors elaborate on the adequacy of this choice?

- The authors aim at minimizing compliance / strain energy, meaning they are aiming at the stiffest cell possible that respects the optimization requirements. Can the authors elaborate if for a tire, which should also fulfil a damping behaviour and is usually not very stiff (air pressure), the optimization function chosen is the most adequate?

- In the document, there is a discussion regarding printing orientation. Can the authors comment why is this necessary? The question would only be relevant if the final product could be printed horizontally and vertically using FFF, but wouldn't a vertically printed wheel be unbalance in the end, and wouldn't the unit cells that compose its structure be radially varying, making the "optimization of cells considering printing orientation" not adequate? Or when the authors mention printing orientation of the unit cells it should be understood as the orientation of the cells within an horizontally printed wheel (if so, what would be the circular shape of the optimization results, e.g. figure 12)?

In Introduction:

Page 2

The sentence “In general, cellular structures can be divided into four categories of foams, lattice, and Triply Periodic Minimal Surfaces (TPMS), which include solid-based TPMs and sheet-based TPMs [3]” seems incorrect since four categories are named but only three defined. Please, clarify.

In the sentence “PTs in general have higher stability and longevity compared to the traditional tires and their fabrication via 3Dprinting will result in energy and cost saving [8-9]”,ccan the authors elaborate on how the stability and longevity of NPTs is greater than traditional tires, and explain if there is no alternative to 3D-printing NPTs? Maybe consider rephrasing to “could result in” or similar in intent.

The acronym “MEW” has not been defined.

When displaying the wheel design by Zhang et al. [33], consider using stiffness instead of hardness. It should be more adequate if the analysis was concerned with the deformation of the structure rather than with mostly a surface deformation phenomenon. Please comment.

Page 3

The sentences “Ju et al. [5] investigated hexagonal honeycomb cells in NPTs. Two types of hexagonal honeycomb are designed: same load carrying capacity and same cell wall thickness.” is not clear. Also concerning what follows “The results show that cells with a very positive angle under the same vertical bearing capacity have low local stresses and low mass” it is not clear what are positive or negative angles. Could the authors maybe clarify what is meant with cells being characterized by low mass?

Authors have yet to explain fundamental concepts to say, “Using the overhang control in connection with 3D printing of non-pneumatic tires, so that there was no need for the support structure, and it provided better bending and compressive properties”. Please consider rephrasing to a more general and comprehensive bullet at this stage of the article.

Page 4: Where is table 1? Starts with table 2...

Page 5:

Authors state "According to this figure, only cells that are at least 3 times the size of the selected cell can successfully be fabricated via FDM 3D printing" whish might be true for the manufacturing parameters and conditions used in this study, but is not necessarily in the case of other works.

The suggestion is to make that clear.

Page 7

It is not clear what "L, and L0" in Figure 5 represent. Please justify.

Page 9

Even though the condition of the overhang angle is a known phenomenon in FDM, it is not clear what Figure 7 wants to represent...Please clarify.

Page 10

It is not clear what is the meaning of the stress values shown in Figure 8, ... Please clarify.

Page 11

If the tire alternates its stress state between bending / compression, and the ratio of -1 used in the fatigue tests. The effect of average stress was considered? A comment would be apreciated.

Page 14

Please verify the units mentioned in "The results of the optimization of the cylindrical unit cell with a diameter and height of 1 m"

Page 17

Please clarify the analysis of the results as they are not easily understood in their current form (1st paragraph).

Page 22

In section 3.2 authors state that "As mentioned in the third chapter, the sample is closed by 6 screws at a distance of one centimetre from the beginning and the end of the sample." But there is no other place in the document where screws are mentioned. Moreover, can the authors clarify the meaning of "Therefore, this part of the sample is considered for closing fixed screws."

Page 26

Can the authors clarify what is the meaning of a “0.7 lifetime” in the sentence “According to the results, in general, it can be said that the normal lifetime of TO and SC samples is the same on average and is around 0.7.”

Images of Table 21 are not with a proper resolution.

The sentence “These samples are designed for use in car tires. Manufacturers are always looking for lower weight in the car. Because the lighter car weight brings benefits such as reducing fuel consumption and emissions. Therefore, if the designer is looking for lightweight design of non-pneumatic tires, he can use this type of cell.”

This is a clear example of a paragraph that need to be English revised. Also, this sentence should be relocated, potentially to the final conclusions or introduction, since similar content in these sections of the document already exist.

Page 31

Figure 23 could be improved. “Defect caused by manufacturing method” is not a conclusive cause of defects. It is not clear what was the printing orientation of the specimens. Please add a comment, and characterize the different defects.

Page 33

Could the authors verify the first bullet according with a previous comment? Also, the same bullet says 4.2 mm, is this the same conclusion of figure 1?

6. PLOS authors have the option to publish the peer review history of their article (what does this mean?). If published, this will include your full peer review and any attached files.

Reviewer #1: No

Reviewer #2: No

---

## [Author Response · Author response to Decision Letter 0]

7 Jul 2023

Please check the file including the answers to comments.

---

## [Decision Letter · Decision Letter 1]

7 Aug 2023

Topology Optimization on Metamaterial Cells for Replacement Possibility in Non-Pneumatic Tire and the Capability of 3D-printing

PONE-D-23-07715R1

Dear Dr. Azadi,

We’re pleased to inform you that your manuscript has been judged scientifically suitable for publication and will be formally accepted for publication once it meets all outstanding technical requirements.

Kind regards,

Antonio Riveiro Rodríguez, PhD

Academic Editor

PLOS ONE

Reviewers' comments:

Reviewer's Responses to Questions

**Comments to the Author**

1. If the authors have adequately addressed your comments raised in a previous round of review and you feel that this manuscript is now acceptable for publication, you may indicate that here to bypass the “Comments to the Author” section, enter your conflict of interest statement in the “Confidential to Editor” section, and submit your "Accept" recommendation.

Reviewer #2: All comments have been addressed

2. Is the manuscript technically sound, and do the data support the conclusions?

Reviewer #2: Yes

3. Has the statistical analysis been performed appropriately and rigorously? 

Reviewer #2: N/A

4. Have the authors made all data underlying the findings in their manuscript fully available?

Reviewer #2: Yes

5. Is the manuscript presented in an intelligible fashion and written in standard English?

Reviewer #2: Yes

6. Review Comments to the Author

Reviewer #2: The authors answered the questions assertively. The paper has been improved and is more robust, although the reviewer thinks it is quite long.

7. PLOS authors have the option to publish the peer review history of their article (what does this mean?). If published, this will include your full peer review and any attached files.

Reviewer #2: No

---

## [Editor Report · Acceptance letter]

22 Aug 2023

PONE-D-23-07715R1 

Topology Optimization on Metamaterial Cells for Replacement Possibility in Non-Pneumatic Tire and the Capability of 3D-printing 

Dear Dr. Azadi:

I'm pleased to inform you that your manuscript has been deemed suitable for publication in PLOS ONE. Congratulations! Your manuscript is now with our production department. 

Kind regards, 

on behalf of

Dr. Antonio Riveiro Rodríguez 

Academic Editor

PLOS ONE